# Near-Optimal Sample Complexity Bounds for Constrained Average-Reward MDPs

**Yukuan Wei**
School of Mathematical Sciences
Fudan University
22300180145@m.fudan.edu.cn

**Xudong Li**
School of Data Science
Fudan University
lixudong@fudan.edu.cn

**Lin F. Yang**
Electrical and Computer Engineering / Computer Science
University of California, Los Angeles
linyang@ee.ucla.edu

## Abstract

Recent advances have significantly improved our understanding of the sample complexity of learning in average-reward Markov decision processes (AMDPs) under the generative model. However, much less is known about the constrained average-reward MDP (CAMDP), where policies must satisfy long-run average constraints. In this work, we address this gap by studying the sample complexity of learning an $\varepsilon$-optimal policy in CAMDPs under a generative model. We propose a model-based algorithm that operates under two settings: (i) *relaxed feasibility*, which allows small constraint violations, and (ii) *strict feasibility*, where the output policy satisfies the constraint. We show that our algorithm achieves sample complexities of $\tilde{O}\left(\frac{SA(B+H)}{\varepsilon^2}\right)$ and $\tilde{O}\left(\frac{SA(B+H)}{\varepsilon^2\zeta^2}\right)$ under the relaxed and strict feasibility settings, respectively. Here, $\zeta$ is the Slater constant indicating the size of the feasible region, $H$ is the span bound of the bias function, and $B$ is the transient time bound. Moreover, a matching lower bound of $\tilde{\Omega}\left(\frac{SA(B+H)}{\varepsilon^2\zeta^2}\right)$ for the strict feasibility case is established, thus providing the first *minimax-optimal* bounds for CAMDPs. Our results close the theoretical gap in understanding the complexity of constrained average-reward MDPs.

## 1 Introduction

Reinforcement learning (RL) (Sutton & Barto, 1998) provides a powerful framework for sequential decision-making under uncertainty, with successes in domains such as game playing (Mnih et al., 2015; Silver et al., 2016), robotic control (Tan et al., 2018; Zeng et al., 2020), clinical decision-making (Schaefer et al., 2005), and aligning large language models with human preferences (Shao et al., 2024; Ouyang et al., 2022). Most classical RL algorithms optimize a single reward signal without additional constraints. Yet in many high-stakes applications, agents must also be safe, fair, and resource-aware, in addition to being efficient in practice. This leads to *constrained Markov decision processes* (CMDPs) (Altman, 1999), which maximize expected reward subject to an auxiliary cost constraint. A representative example arises in wireless sensor networks (Buratti et al., 2009; Julian et al., 2002), balancing high throughput with average power constraints.

A growing body of work investigates constrained reinforcement learning in unknown environments (Efroni et al., 2020; Zheng & Ratliff, 2020; Qiu et al., 2020; Brantley et al., 2020; Kalagarla et al., 2021; Yu et al., 2021; Ding et al., 2021; Gattami et al., 2021; Miryoosefi & Jin, 2022). These efforts focus on the online learning setting, aiming to minimize both regret and constraint violation under exploration, estimation, and policy optimization challenges in finite-state, finite-action CMDPs. In contrast, recent work (HasanzadeZonuzy et al., 2021; Wei et al., 2021; Bai et al., 2021; Vaswani et al., 2022) studies the *generative model* (Kearns & Singh, 1999; Kakade, 2003; Agarwal et al.,

2020; Sidford et al., 2018; Yang & Wang, 2019), i.e., a simulator that provides sample transitions and rewards for any queried state-action pair, removing the need for exploration.

Most prior work centers on finite-horizon or discounted MDPs, where either the horizon is fixed to $T$ steps or future rewards are geometrically discounted by $\gamma^t$. While analytically convenient, these formulations can limit long-term performance: finite-horizon methods impose a cutoff, while discounting attenuates future rewards. To address this, the *average-reward MDP* (AMDP) framework (Puterman, 2014) seeks to maximize steady-state long-run average reward.

Although planning in AMDPs is relatively well-understood (Altman, 1999; Borkar, 2005; Borkar & Jain, 2014), sample complexity for learning $\varepsilon$-optimal policies was challenging due to the lack of episode resets and the need to reason about long-term behavior without discounting. Recent advances in the generative model setting establish near-optimal bounds depending on the *optimal bias span $H$* and the *mixing or transient time $B$*, yielding rates $\tilde{\Theta}\left(\frac{SA(B+H)}{\varepsilon^2}\right)$ (Zurek & Chen, 2024), where $S$ is the number of states and $A$ the number of actions.

Despite this progress for unconstrained AMDPs, the constrained variant–*constrained average-reward MDPs* (CAMDPs)–remains poorly understood. In CAMDPs, the agent must maximize steady-state average reward while satisfying an average constraint on cost, risk, or resource usage, capturing long-term fairness, sustainability, and safe deployment. While discounted CMDPs have seen progress in both relaxed and strict feasibility regimes (Vaswani et al., 2022), there are still no known sample complexity bounds for learning in CAMDPs, and the fundamental statistical limits in relaxed or strict feasibility settings remain open.

This gap motivates our work. We initiate the study of the sample complexity of learning $\varepsilon$-optimal policies in CAMDPs under the generative model. We develop a model-based primal-dual algorithm handling both *relaxed feasibility*, where the returned policy may violate the constraint by at most $\varepsilon$, and *strict feasibility*, where the policy must satisfy it exactly. We establish matching near-optimal upper and lower bounds in terms of the bias span, the transient time, and the *Slater constant $\zeta$*. While relaxed and strict feasibility have been studied in discounted CMDPs (Vaswani et al., 2022), our work provides the first sample complexity characterization for CAMDPs in the average-reward setting. Below, we summarize our contributions in more detail.

**Our contributions.** We present the first near-optimal sample-complexity bounds for learning in CAMDPs with access to a generative model:

• We design a model-based algorithm that returns an $\varepsilon$-optimal policy for CAMDPs under both relaxed and strict feasibility. Our method relies on solving a sequence of unconstrained average-reward MDPs using black-box planners.

• In the relaxed feasibility setting, we prove that our algorithm requires at most $\tilde{O}\left(\frac{SA(B+H)}{\varepsilon^2}\right)$ samples, where $S$ and $A$ are the number of states and actions, $H$ is the span bound of the bias function, $B$ is a transient time bound, and $\zeta$ is the *Slater constant* characterizing the size of the feasible region.

• In the strict feasibility setting, the sample complexity increases to $\tilde{O}\left(\frac{SA(B+H)}{\varepsilon^2\zeta^2}\right)$, and we show that this dependence on $\zeta$ is necessary by proving a matching lower bound of $\tilde{\Omega}\left(\frac{SA(B+H)}{\varepsilon^2\zeta^2}\right)$. These are the first lower bounds for strict feasibility in CAMDPs, establishing a provable separation between the relaxed and strict regimes.

Together, our results provide the first near *minimax-optimal* sample complexity bounds for constrained average-reward reinforcement learning with respect to $S$, $A$, $B$ and $H$ and reveal fundamental insights into how long-run constraints affect the hardness of planning under uncertainty.

## 1.1 RELATED WORKS

There is a large body of research on the sample complexity of learning in *unconstrained* Markov decision processes (MDPs); see the monograph by Agarwal et al. (2019) for a comprehensive overview. In parallel, substantial progress has been made in *constrained* reinforcement learning under unknown dynamics (Efroni et al., 2020; Zheng & Ratliff, 2020; Qiu et al., 2020; Brantley et al., 2020; Kalagarla et al., 2021; Yu et al., 2021; Ding et al., 2021; Gattami et al., 2021; Miryoosefi & Jin, 2022), particularly in finite-horizon settings. Another line of work addresses *discounted* constrained MDPs

(CMDPs) with access to a generative model (HasanzadeZonuzy et al., 2021; Wei et al., 2021; Bai et al., 2021; Vaswani et al., 2022), yielding sample-efficient algorithms under both relaxed and strict constraint satisfaction.

In contrast, the average-reward setting is less explored. For unconstrained average-reward MDPs, Zurek & Chen (2024) established nearly minimax-optimal bounds under a generative model, showing that $\widetilde{O}\left(SAH/\varepsilon^2\right)$ samples suffice for weakly communicating MDPs, where $H$ is the span of the optimal bias function. They further introduced a transient time parameter $B$ to handle general multichain MDPs, proving a matching bound of $\widetilde{O}\left(SA(B+H)/\varepsilon^2\right)$. However, their analysis does not incorporate constraints, and extending their framework to constrained average-reward MDPs (CAMDPs) remains open.

Among works on CMDPs, Vaswani et al. (2022) provided the first minimax-optimal sample complexity bounds for the *discounted* setting via dual linear programming. Yet their techniques do not extend to average-reward problems, where key properties like Bellman contraction no longer hold. In a separate effort, Bai et al. (2024) studied CAMDPs in an online model-free setting with general policy classes, establishing sublinear regret for constraint violation and the duality gap. Their results, however, focus on asymptotic behavior and do not provide near-optimal finite-sample guarantees under a generative model.

To our knowledge, this work is the first to establish near-optimal sample complexity bounds for CAMDPs under both relaxed and strict feasibility in the generative model setting. We propose a primal-dual algorithm that achieves minimax-optimal rates in terms of the number of states, actions, bias span, transient time, and the Slater constant, thereby unifying and extending existing results from both the unconstrained and discounted settings.

## 2 PROBLEM FORMULATION AND PRELIMINARIES

We study an infinite-horizon constrained average-reward Markov decision process (CAMDP), denoted by $M$ and specified by the tuple $\langle \mathcal{S}, \mathcal{A}, \mathcal{P}, r, c, b, s \rangle$. Here, $\mathcal{S}$ and $\mathcal{A}$ denote the sets of states and actions; $\mathcal{P} : \mathcal{S} \times \mathcal{A} \to \Delta_{\mathcal{S}}$ is the transition probability kernel; and $s \in \Delta_{\mathcal{S}}$ represents the initial state distribution. The objective is to maximize the primary reward function $r : \mathcal{S} \times \mathcal{A} \to [0, 1]$, subject to a constraint $c : \mathcal{S} \times \mathcal{A} \to [0, 1]$. Given that $\Delta_{\mathcal{A}}$ denotes the probability simplex over actions, the expected average reward under a stochastic stationary policy $\pi : \mathcal{S} \to \Delta_{\mathcal{A}}$ is defined as $\rho_r^\pi(s) = \lim_{T \to \infty} \frac{1}{T} \mathbb{E}_s\left[\sum_{t=0}^{T-1} r(s_t, a_t)\right]$, where $s_0 \sim s$, $a_t \sim \pi(\cdot \mid s_t)$, and $s_{t+1} \sim \mathcal{P}(\cdot \mid s_t, a_t)$. The *bias function* of a stationary policy $\pi$ is $h_r^\pi(s) := \text{C-}\lim_{T \to \infty} \mathbb{E}_s^\pi\left[\sum_{t=0}^{T-1}(r_t - \rho_r^\pi(S_t))\right]$, where C-lim denotes the Cesàro limit. When the Markov chain induced by $P_\pi$ is aperiodic, the Cesàro limit coincides with the standard limit. For any policy $\pi$, the pair $(\rho^\pi, h^\pi)$ satisfies the Bellman-like relations $\rho_r^\pi = P_\pi \rho_r^\pi$ and $\rho_r^\pi + h_r^\pi = r_\pi + P_\pi h_r^\pi$. Similarly, define the *constraint value function* and *constraint bias function* of $\pi$ as $\rho_c^\pi$ and $h_c^\pi$. The objective in a CAMDP is to find a policy solving the following optimization problem:

$$\max_\pi \rho_r^\pi(s) \quad \text{s.t.} \quad \rho_c^\pi(s) \geq b. \tag{1}$$

We denote the optimal stochastic policy by $\pi^*$, and its corresponding reward value by $\rho_r^*(s)$.

**Weakly communicating setting** A Markov decision process (MDP) is *weakly communicating* if its state space $\mathcal{S}$ can be partitioned into two disjoint subsets $\mathcal{S} = \mathcal{S}_1 \cup \mathcal{S}_2$, such that all states in $\mathcal{S}_1$ are transient under any stationary policy, and for any $s, s' \in \mathcal{S}_2$ there exists a stationary policy making $s'$ reachable from $s$. In such MDPs the average reward vector $\rho^*$ is constant, i.e., $\rho^*(s) = \rho^*$ for all $s \in \mathcal{S}$. Consequently, $(\rho^*, h^*)$ satisfies the *average-reward optimality equation*:

$$\rho^* + h^*(s) = \max_{a \in \mathcal{A}}\{r(s, a) + \sum_{s'} P(s' \mid s, a)h^*(s')\}, \quad \forall s \in \mathcal{S}.$$

We occasionally abuse notation and treat $\rho^*$ as a scalar. A stationary policy is *multichain* if it induces multiple closed irreducible recurrent classes, and an MDP is *multichain* if it admits at least one such policy. While general MDPs may only possess multichain gain-optimal policies with non-constant $\rho^*$, any weakly communicating MDP admits at least one unichain gain-optimal policy under which $\rho^*$ is uniform. Moreover, every *uniformly mixing* MDP is weakly communicating. A stronger assumption is *communicating*, which excludes transient states and requires every state be reachable from every other under every stationary policy.

**Complexity parameters** We introduce several problem-dependent parameters characterizing the complexity of constrained average-reward MDPs. The diameter $D$ is $D := \max_{s_1 \neq s_2} \min_\pi \mathbb{E}_{s_1}^\pi [\tau_{s_2}]$, where $\tau_{s_2}$ is the first hitting time to $s_2$ under $\pi$. The span bound of the bias function is $H := \max_\pi \|h\|_{\text{span}}$ with $\|v\|_{\text{span}} := \max_s v(s) - \min_s v(s)$, capturing cumulative reward range and long-term difficulty. We also introduce the *transient time parameter* $B$. Let $\Pi$ be the set of stationary deterministic policies. For $\pi \in \Pi$, define recurrent states $\mathcal{R}^\pi$ and transient states $\mathcal{T}^\pi = \mathcal{S} \setminus \mathcal{R}^\pi$ under $P_\pi$, and let $T_{\mathcal{R}^\pi} = \inf\{t \geq 0 : S_t \in \mathcal{R}^\pi\}$ be the first hitting time to a recurrent state. An MDP satisfies the *bounded transient time property* with parameter $B$ if $\mathbb{E}_s^\pi [T_{\mathcal{R}^\pi}] \leq B$ for all $\pi \in \Pi$ and $s \in \mathcal{S}$, ensuring uniformly bounded time in transient states. Finally, the *Slater constant* is $\zeta := \max_\pi \rho_c^\pi(s) - b$ (Ding et al., 2021; Bai et al., 2021), measuring the feasibility margin and how difficult it is to satisfy the constraint.

**Blackwell-optimal policy** A policy $\pi^*$ is Blackwell-optimal if there exists some discount factor $\bar{\gamma} \in (0, 1)$ such that for all $\gamma \geq \bar{\gamma}$ we have $V_\gamma^{\pi^*} \geq V_\gamma^\pi$ for all policies $\pi$. Henceforth we let $\pi^*$ denote some fixed Blackwell-optimal policy, which is guaranteed to exist when $S$ and $A$ are finite (Puterman, 2014). We define the optimal gain $\rho^* \in \mathbb{R}^\mathcal{S}$ by $\rho^*(s) = \sup_\pi \rho^\pi(s)$ and note that we have $\rho^* = \rho^{\pi^*}$. For all $s \in \mathcal{S}$, $\rho^*(s) \geq \max_{a \in \mathcal{A}} P_{sa} \rho^*$, or equivalently $\rho^\star(s) \geq P_\pi \rho^*$ for all policies $\pi$ (and this maximum is achieved by $\pi^*$). We also define $h^* = h^{\pi^*}$ (and we note that this definition does not depend on which Blackwell-optimal $\pi^*$ is used, if there are multiple). For all $s \in \mathcal{S}$, $\rho^*$ and $h^*$ satisfy $\rho^*r(s) + h^*(s) = \max_{a \in \mathcal{A}: P_{sa}\rho^* = \rho^*(s)} r_{sa} + P_{sa} h^*$, known as the (unmodified) Bellman equation.

**Learning framework** For clarity of exposition, we assume that the reward functions $r$ and $c$ are known, while the transition dynamics $\mathcal{P}$ are unknown and must be learned. This assumption does not affect the leading-order sample complexity, as estimating rewards is generally easier than estimating the transition matrix (Azar et al., 2013; Sidford et al., 2018). We further assume access to a *generative model* (simulator), which allows the agent to draw samples from $\mathcal{P}(\cdot \mid s, a)$ for any state-action pair $(s, a)$. Under this setting, our objective is to characterize the sample complexity required to compute an approximately optimal policy $\hat{\pi}$ for the CAMDP $M$. Given a desired accuracy level $\varepsilon > 0$, we consider two distinct notions of settings:

**Relaxed feasibility** We require the returned policy $\hat{\pi}$ to achieve near-optimal reward, allowing for a small violation of the constraint. Formally, we seek $\hat{\pi}$ such that:

$$\rho_r^{\hat{\pi}}(s) \geq \rho_r^*(s) - \varepsilon, \quad \text{and} \quad \rho_c^{\hat{\pi}}(s) \geq b - \varepsilon. \tag{2}$$

**Strict feasibility** We require $\hat{\pi}$ to achieve near-optimal reward while exactly satisfying the constraint, i.e., zero constraint violation:

$$\rho_r^{\hat{\pi}}(s) \geq \rho_r^*(s) - \varepsilon, \quad \text{and} \quad \rho_c^{\hat{\pi}}(s) \geq b. \tag{3}$$

In the following sections, we describe a general model-based algorithm that can handle both the relaxed and strict feasibility settings, and we instantiate it appropriately for each case.

## 3 METHODOLOGY

We will use a model-based approach for achieving the objectives in Eq. (2) and Eq. (3). In particular, for each $(s, a)$ pair, we collect $N$ independent samples from $\mathcal{P}(\cdot|s, a)$ and form an empirical transition matrix $\hat{\mathcal{P}}$ such that $\hat{\mathcal{P}}(s'|s, a) = \frac{N(s'|s,a)}{N}$, where $N(s'|s, a)$ is the number of samples that have transitions from $(s, a)$ to $s'$. These estimated transition probabilities are used to form a series of empirical discounted MDPs, the result of which will be used as the near optimal solution for a series of corresponding AMDPs. In particular, for each $s \in \mathcal{S}$ and $a \in \mathcal{A}$, we define the perturbed rewards $r_p(s, a) := r(s, a) + Z(s, a)$ where $Z(s, a) \sim \mathcal{U}[0, \omega]$ are i.i.d. uniform random variables and we set other parameters, such as $\bar{\varepsilon} = B + H$, $\gamma = 1 - \frac{\varepsilon_{opt}}{4\bar{\varepsilon}}$ and $\omega = (1 - \gamma)\bar{\varepsilon}/6$ to specify the empirical AMDPs. Finally, compared to Eq. (1), we will require solving the CAMDP with a constraint right-hand side equal to $b'$. Note that setting $b' < b$ corresponds to loosening the constraint, while $b' > b$ corresponds to tightening the constraint. This completes the specification of a series of empirical AMDPs $\{\hat{M}_t\}$ that are defined by the tuple $\langle \mathcal{S}, \mathcal{A}, \hat{\mathcal{P}}, r_p + \lambda_t c, s \rangle$. Furthermore, we will compute the optimal policy for the empirical CAMDP $\hat{M}$ introduced by the generative model as follows:

$$\hat{\pi}^* \in \arg\max \hat{\rho}_{r_p}^\pi(s) \text{ s.t. } \hat{\rho}_c^\pi(s) \geq b' \tag{4}$$

---

**Algorithm 1: Model-based Algorithm for CAMDPs with Generative Model**

---

1 **Input:** $\mathcal{S}$ (state space), $\mathcal{A}$ (action space), $r$ (rewards), $c$ (constraint rewards), $\zeta$ (Slater constant), $N$ (number of samples), $b'$ (constraint RHS), $U$ (projection upper bound),$\varepsilon_1$ (epsilon-net resolution), $T$ (number of iterations), $\lambda_0 = 0$ (initialization), $\varepsilon_{\text{opt}}$ (target accuracy), $\gamma$ (discount factor).

2 For each $(s,a) \in \mathcal{S} \times \mathcal{A}$, collect $n$ samples $S_{s,a}^1, \ldots, S_{s,a}^n$ from $\mathcal{P}(\cdot|s,a)$

3 Form $\hat{\mathcal{P}}$: $\hat{\mathcal{P}}(s'|s,a) = \frac{1}{N} \sum_{i=1}^n \mathbf{1}\{S_{s,a}^i = s'\}, \quad \forall s' \in \mathcal{S}.$

4 Set discount factor $\gamma = 1 - \frac{\varepsilon_{\text{opt}}}{4(B+H)}$

5 Perturb the rewards to form $r_p(s,a) = r(s,a) + Z(s,a)$ where $Z(s,a) \sim \text{Unif}(0,\omega)$.

6 Form the epsilon-net $\Lambda = \{0, \varepsilon_1, 2\varepsilon_1, \ldots, U\}$.

7 **for** $t \leftarrow 0$ **to** $T-1$ **do**

8     Update the Blackwell-optimal policy $\hat{\pi}_t$ by solving the empirical unconstrained AMDP $(\hat{P}, r_p + \lambda_t c)$.

9     Update the dual variable:$\lambda_{t+1} = \mathcal{R}_\Lambda \left[ \mathbb{P}_{[0,U]} \left[ \lambda_t - \eta \left( \rho_c^{\hat{\pi}_t}(s) - b' \right) \right] \right].$

10 **end for**

11 Output the mixture policy:$\hat{\pi} = \frac{1}{T} \sum_{t=0}^{T-1} \hat{\pi}_t.$

---

We will require solving Eq. (4) using a specific primal-dual approach that we outline next. Using this algorithm enables us to prove optimal sample complexity bounds under both relaxed and strict feasibility.

First, observe that Eq. (4) can be written as an equivalent saddle-point problem – $\max_\pi \min_{\lambda \geq 0} [\rho_r^\pi(s) + \lambda (\rho_c^\pi(s) - b')]$, where $\lambda \in \mathbb{R}$ corresponds to the Lagrange multiplier for the constraint. The solution to this saddle-point problem is $(\hat{\pi}^*, \lambda^*)$ where $\hat{\pi}^*$ is the optimal policy for $M'$ and $\lambda^*$ is the optimal Lagrange multiplier. We solve the above saddle-point problem iteratively, by alternatively updating the policy (primal variable) and the Lagrange multiplier (dual variable). If $T$ is the total number of iterations of the primal-dual algorithm, we define $\hat{\pi}_t$ and $\lambda_t$ to be the primal and dual iterates for $t \in [T] := \{1, \ldots, T\}$. The primal update at iteration $t$ is given as:

$$\hat{\pi}_t = \arg\max \left[ \rho_{r_p}^\pi + \lambda_t \rho_c^\pi \right] = \arg\max \rho_t^\pi. \tag{5}$$

Hence, iteration $t$ of the algorithm requires solving an unconstrained MDP with a reward equal to $r_p + \lambda_t c$. This can be done using any black-box MDP solver such as policy iteration. The algorithm updates the Lagrange multipliers using a gradient descent step and requires projecting. In particular, the dual variables are projected onto the $[0, U]$ interval, where $U$ is chosen to be an upper-bound on $|\lambda^*|$.

The dual update at iteration $t$ is given as:

$$\lambda_{t+1} = \mathcal{R}_\Lambda \left[ \mathbb{P}_{[0,U]} \left[ \lambda_t - \eta \left( \rho_c^{\hat{\pi}_t}(s) - b' \right) \right] \right], \tag{6}$$

where $\mathbb{P}_{[0,U]}[\lambda] = \arg\min_{p \in [0,U]} |\lambda - p|$ projects $\lambda$ onto the $[0, U]$ interval. Finally, $\eta$ in Eq. (6) corresponds to the step-size for the gradient descent update. The above primal-dual updates are similar to the dual-descent algorithm proposed in Vaswani et al. (Vaswani et al., 2022). The pseudo-code summarizing the entire model-based algorithm is given in Algorithm 1. We note that although Algorithm 1 requires the knowledge of $\zeta$, this is not essential and we can instead use an estimate of $\zeta$. Next, we show that the primal-dual updates in Algorithm 1 can be used to solve a reference CAMDP. Specifically, we prove the following theorem that bounds the average optimality gap (in the reward value function) and constraint violation for the mixture policy returned by Algorithm 1.

**Theorem 1** (Guarantees for the primal-dual algorithm). For a target error $\varepsilon_{\text{opt}} > 0$, consider the primal-dual updates given in Eq. (5)–Eq. (6) with parameters $U > |\lambda^*|$, $T = \frac{U^2}{\varepsilon_{\text{opt}}^2} \left[ 1 + \frac{1}{(U-\lambda^*)^2} \right]$, $\varepsilon_1 = \frac{\varepsilon_{\text{opt}}^2 (U-\lambda^*)}{6U}$ and $\eta = \frac{U}{\sqrt{T}}$, then the resulting mixture policy $\hat{\pi} := \frac{1}{T} \sum_{t=0}^{T-1} \hat{\pi}_t$ satisfies

$$\rho_{r_p}^{\hat{\pi}}(s) \geq \rho_{r_p}^{\hat{\pi}^*}(s) - \varepsilon_{\text{opt}} \quad \text{and} \quad \rho_c^{\hat{\pi}}(s) \geq b' - \varepsilon_{\text{opt}}.$$

Hence, with $T = O(1/\varepsilon_{\text{opt}}^2)$, the algorithm outputs a policy $\hat{\pi}$ that achieves a reward $\varepsilon_{\text{opt}}$ close to that of the optimal empirical policy $\hat{\pi}^*$, while violating the constraint by at most $\varepsilon_{\text{opt}}$. Hence, with a sufficient number of iterations $T$, we can use the above primal-dual algorithm to approximately solve the problem in Eq. (4). In order to completely instantiate the primal-dual algorithm, we require setting $U > |\lambda^*|$. We will subsequently do this for the relaxed and strict feasibility settings in Section 4.

## 4  UPPER-BOUND UNDER RELAXED FEASIBILITY

In order to achieve the objective in Eq. (2) for a target error $\varepsilon > 0$, we require setting $N = \tilde{O}\left(\frac{SA(B+H)}{\varepsilon^2}\right)$, $b' = b - \frac{3\varepsilon}{8}$ and $\omega = \frac{\varepsilon(1-\gamma)}{8}$. This completely specifies the empirical CMDP $\hat{M}$ and the problem in Eq. (4). In order to specify the primal-dual algorithm, we set $U = O(1/\varepsilon(1-\gamma))$, $\varepsilon_1 = O\left(\varepsilon^2(1-\gamma)^2\right)$, $T = O(1/(1-\gamma)^4\varepsilon^4)$ and $\gamma = 1 - \frac{\varepsilon_{\text{opt}}}{4(B+H)}$. With these choices, we prove the following theorem in Appendix B and provide a proof sketch below.

**Theorem 2.** For a fixed $\varepsilon \in (0,1]$, $\delta \in (0,1)$ and a general CAMDP, suppose the corresponding AMDPs $(\mathcal{P}, r)$ and $(\mathcal{P}, c)$ have bias functions bound $H$, and satisfy the bounded transient time assumption with parameter $B$. Algorithm 1 with $N = \tilde{O}\left(\frac{SA(B+H)}{\varepsilon^2}\right)$ samples, $b' = b - \frac{3\varepsilon}{8}$, $\omega = \frac{\varepsilon(1-\gamma)}{8}$, $U = O(1/\varepsilon(1-\gamma))$, $\varepsilon_1 = O\left(\varepsilon^2(1-\gamma)^2\right)$, $T = O(1/(1-\gamma)^4\varepsilon^4)$ and $\gamma = 1 - \frac{\varepsilon_{\text{opt}}}{4(B+H)}$, returns policy $\hat{\pi}$ that satisfies the objective in Eq. (2) with probability at least $1 - 4\delta$.

*Proof Sketch:* We prove the result for a general primal-dual error $\varepsilon_{\text{opt}} < \varepsilon$ and $b' = b - \frac{\varepsilon - \varepsilon_{\text{opt}}}{2}$, and subsequently specify $\varepsilon_{\text{opt}}$ and hence $b'$. In Lemma 9 (proved in Appendix B), we show that if the constraint value functions are sufficiently concentrated (the empirical value function is close to the ground truth value function) for both the optimal policy $\pi^*$ in $M$ and the mixture policy $\hat{\pi}$ returned by Algorithm 1, i.e., if

$$\left|\rho_c^{\hat{\pi}}(s) - \hat{\rho}_c^{\hat{\pi}}(s)\right| \leq \frac{\varepsilon - \varepsilon_{\text{opt}}}{2} \;;\; \left|\rho_c^{\pi^*}(s) - \hat{\rho}_c^{\pi^*}(s)\right| \leq \frac{\varepsilon - \varepsilon_{\text{opt}}}{2}, \tag{7}$$

then (i) policy $\hat{\pi}$ violates the constraint in $M$ by at most $\varepsilon$, i.e., $\rho_c^{\hat{\pi}}(s) \geq b - \varepsilon$, and (ii) its suboptimality in $M$ (compared to $\pi^*$) can be decomposed as:

$$\rho_r^{\pi^*}(s) - \rho_r^{\hat{\pi}}(s) \leq 2\omega + \varepsilon_{\text{opt}} + \left|\rho_{r_p}^{\pi^*}(s) - \hat{\rho}_{r_p}^{\pi^*}(s)\right| + \left|\hat{\rho}_{r_p}^{\hat{\pi}}(s) - \rho_{r_p}^{\hat{\pi}}(s)\right| \tag{8}$$

In order to instantiate the primal-dual algorithm, we require a concentration result for policy $\pi_c^*$ that maximizes the constraint value function, i.e. if $\pi_c^* := \arg\max \rho_c^\pi(s)$, then we require $\left|\hat{\rho}_c^{\pi_c^*} - \rho_c^{\pi_c^*}(s)\right| \leq \varepsilon + \varepsilon_{\text{opt}}$. In Case 1 of Lemma 6 (proved in Appendix A), we show that if this concentration result holds, then we can upper-bound the optimal dual variable $|\lambda^*|$ by $\frac{2(1+\omega)}{(\varepsilon + \varepsilon_{\text{opt}})}$. With these results in hand, we can instantiate all the algorithm parameters except $N$ (the number of samples required for each state-action pair). In particular, we set $\varepsilon_{\text{opt}} = \frac{\varepsilon}{4}$ and hence $b' = b - \frac{3\varepsilon}{8}$, and $\omega = \frac{\varepsilon(1-\gamma)}{8} < 1$. Setting $U = \frac{32}{5\varepsilon(1-\gamma)}$ ensures that the $U > |\lambda^*|$ condition required by Theorem 1 holds. To guarantee that the primal-dual algorithm outputs an $\frac{\varepsilon}{4}$-approximate policy, we use Theorem 1 to set $T = O\left(\frac{1}{(1-\gamma)^4\varepsilon^4}\right)$ iterations and $\varepsilon_1 = O\left(\varepsilon^2(1-\gamma)^2\right)$. Eq. (8) can then be simplified as,

$$\rho_r^{\pi^*}(s) - \rho_r^{\hat{\pi}}(s) \leq \frac{\varepsilon}{2} + \left|\rho_{r_p}^{\pi^*}(s) - \hat{\rho}_{r_p}^{\pi^*}(s)\right| + \left|\hat{\rho}_{r_p}^{\hat{\pi}}(s) - \rho_{r_p}^{\hat{\pi}}(s)\right|.$$

Putting everything together, in order to guarantee an $\varepsilon$-reward suboptimality for $\hat{\pi}$, we require that:

$$\left|\hat{\rho}_c^{\pi_c^*} - \rho_c^{\pi_c^*}(s)\right| \leq \frac{5\varepsilon}{4} \;;\; \left|\rho_c^{\hat{\pi}}(s) - \hat{\rho}_c^{\hat{\pi}}(s)\right| \leq \frac{3\varepsilon}{8} \;;\; \left|\rho_c^{\pi^*}(s) - \hat{\rho}_c^{\pi^*}(s)\right| \leq \frac{3\varepsilon}{8}$$

$$\left|\rho_{r_p}^{\pi^*}(s) - \hat{\rho}_{r_p}^{\pi^*}(s)\right| \leq \frac{\varepsilon}{4} \;;\; \left|\hat{\rho}_{r_p}^{\hat{\pi}}(s) - \rho_{r_p}^{\hat{\pi}}(s)\right| \leq \frac{\varepsilon}{4}. \tag{9}$$

We control such concentration terms for both the constraint and reward value functions in Appendix B, and bound the terms in Eq. (9). In particular, we prove that for a fixed $\varepsilon \in (0, 1/1-\gamma]$, using

$N \geq \tilde{O}\left(\frac{SA(B+H)}{\varepsilon^2}\right)$ samples ensures that the statements in Eq. (9) hold with probability $1 - 4\delta$. This guarantees that $\rho_r^{\pi^*}(s) - \rho_r^{\hat{\pi}}(s) \leq \varepsilon$ and $\rho_c^{\hat{\pi}}(s) \geq b - \varepsilon$. □

## 5 Upper-bound under Strict Feasibility

Unlike Section 4, since the strict feasibility setting does not allow any constraint violations, it necessitates using a stricter constraint in the empirical CMDP to account for the estimation error in the transition probabilities. Algorithmically, we require setting $b' > b$. Specifically, in order to achieve the objective in Eq. (3) for a target error $\varepsilon > 0$, we require setting $N = \tilde{O}\left(\frac{SA(B+H)}{\varepsilon^2\zeta^2}\right)$, $b' = b + \frac{\varepsilon(1-\gamma)\zeta}{20}$ and $\omega = \frac{\varepsilon(1-\gamma)}{10}$. This completely specifies the empirical CMDP $\hat{M}$ and the problem in Eq. (4). To specify the primal-dual algorithm, we set $U = \frac{4(1+\omega)}{\zeta(1-\gamma)}$, $\varepsilon_1 = O\left(\varepsilon^2(1-\gamma)^4\zeta^2\right)$, $T = O\left(1/(1-\gamma)^6\zeta^4\varepsilon^2\right)$ and $\gamma = 1 - \frac{\varepsilon_{\text{opt}}}{4(B+H)}$. With these choices, we prove the following theorem in Appendix C, and provide a proof sketch below.

> **Theorem 3.** For a fixed $\varepsilon \in (0, 1/1-\gamma]$ and $\delta \in (0, 1)$, Algorithm 1, with $N = \tilde{O}\left(\frac{SA(B+H)}{\varepsilon^2\zeta^2}\right)$ samples, $b' = b + \frac{\varepsilon(1-\gamma)\zeta}{20}$, $\omega = \frac{\varepsilon(1-\gamma)}{10}$, $U = \frac{4(1+\omega)}{\zeta(1-\gamma)}$, $\varepsilon_1 = O\left(\varepsilon^2(1-\gamma)^4\zeta^2\right)$, $T = O\left(1/(1-\gamma)^6\zeta^4\varepsilon^2\right)$ and $\gamma = 1 - \frac{\varepsilon_{\text{opt}}}{4(B+H)}$ returns policy $\hat{\pi}$ that satisfies the objective in Eq. (3), with probability at least $1 - 4\delta$.

*Proof Sketch:* We prove the result for a general $b' = b + \Delta$ for $\Delta > 0$ and primal-dual error $\varepsilon_{\text{opt}} < \Delta$, and subsequently specify $\Delta$ (and hence $b'$) and $\varepsilon_{\text{opt}}$. In Lemma 10 (proved in Appendix C), we prove that if the constraint value functions are sufficiently concentrated (the empirical value function is close to the ground truth value function) for both the optimal policy $\pi^*$ in $M$ and the mixture policy $\hat{\pi}$ returned by Algorithm 1 i.e. if

$$\left|\rho_c^{\hat{\pi}}(s) - \hat{\rho}_c^{\hat{\pi}}(s)\right| \leq \Delta - \varepsilon_{\text{opt}} \quad ; \quad \left|\rho_c^{\pi^*}(s) - \hat{\rho}_c^{\pi^*}(s)\right| \leq \Delta \tag{10}$$

then (i) policy $\hat{\pi}$ satisfies the constraint in $M$ i.e. $\rho_c^{\hat{\pi}}(s) \geq b$, and (ii) its suboptimality in $M$ (compared to $\pi^*$) can be decomposed as:

$$\rho_r^{\pi^*}(s) - \rho_r^{\hat{\pi}}(s) \leq 2\omega + \varepsilon_{\text{opt}} + 2\Delta|\lambda^*| + \left|\rho_{r_p}^{\pi^*}(s) - \hat{\rho}_{r_p}^{\pi^*}(s)\right| + \left|\hat{\rho}_{r_p}^{\hat{\pi}}(s) - \rho_{r_p}^{\hat{\pi}}(s)\right| \tag{11}$$

In order to upper-bound $|\lambda^*|$, we require a concentration result for policy $\pi_c^* := \arg\max \rho_c^\pi(s)$ that maximizes the constraint value function. In particular, we require $\Delta \in \left(0, \frac{\zeta}{2}\right)$ and $\left|\rho_c^{\pi_c^*}(s) - \hat{\rho}_c^{\pi_c^*}(s)\right| \leq \frac{\zeta}{2} - \Delta$. In Case 2 of Lemma 6 (proved in Appendix A), we show that if this concentration result holds, then we can upper-bound the optimal dual variable $|\lambda^*|$ by $\frac{2(1+\omega)}{\zeta(1-\gamma)}$. Using the above bounds to simplify Eq. (11),

$$\rho_r^{\pi^*}(s) - \rho_r^{\hat{\pi}}(s) \leq \frac{2\omega}{1-\gamma} + \varepsilon_{\text{opt}} + \frac{4\Delta(1+\omega)}{\zeta(1-\gamma)} + \left|\rho_{r_p}^{\pi^*}(s) - \hat{\rho}_{r_p}^{\pi^*}(s)\right| + \left|\hat{\rho}_{r_p}^{\hat{\pi}}(s) - \rho_{r_p}^{\hat{\pi}}(s)\right|.$$

With these results in hand, we can instantiate all the algorithm parameters except $N$ (the number of samples required for each state-action pair). In particular, we set $\Delta = \frac{\varepsilon(1-\gamma)\zeta}{40} < \frac{\zeta}{2}$, $\varepsilon_{\text{opt}} = \frac{\Delta}{5} = \frac{\varepsilon(1-\gamma)\zeta}{200} < \frac{\varepsilon}{5}$, and $\omega = \frac{\varepsilon(1-\gamma)}{10} < 1$. We set $U = \frac{8}{\zeta(1-\gamma)}$ for the primal-dual algorithm, ensuring that the $U > |\lambda^*|$ condition required by Theorem 1 holds. In order to guarantee that the primal-dual algorithm outputs an $\frac{\varepsilon(1-\gamma)\zeta}{200}$-approximate policy, we use Theorem 1 to set $T = O\left(\frac{1}{(1-\gamma)^6\zeta^4\varepsilon^2}\right)$ iterations and $\varepsilon_1 = O\left(\varepsilon^2(1-\gamma)^4\zeta^2\right)$. With these values, we can further simplify Eq. (11),

$$\rho_r^{\pi^*}(s) - \rho_r^{\hat{\pi}}(s) \leq \frac{3\varepsilon}{5} + \left|\rho_{r_p}^{\pi^*}(s) - \hat{\rho}_{r_p}^{\pi^*}(s)\right| + \left|\hat{\rho}_{r_p}^{\hat{\pi}}(s) - \rho_{r_p}^{\hat{\pi}}(s)\right|.$$

Putting everything together, in order to guarantee an $\varepsilon$-reward suboptimality for $\hat{\pi}$, we require the following concentration results to hold for $\Delta = \frac{\varepsilon(1-\gamma)\zeta}{40}$,

$$\left|\rho_c^{\hat{\pi}}(s) - \hat{\rho}_c^{\hat{\pi}}(s)\right| \le \frac{4\Delta}{5} \,;\, \left|\rho_c^{\pi^*}(s) - \hat{\rho}_c^{\pi^*}(s)\right| \le \Delta \,;\, \left|\rho_c^{\pi_c^*}(s) - \hat{\rho}_c^{\pi_c^*}(s)\right| \le \frac{19\Delta}{5}$$

$$\left|\rho_{r_p}^{\pi^*}(s) - \hat{\rho}_{r_p}^{\pi^*}(s)\right| \le \frac{\varepsilon}{5} \,;\, \left|\hat{\rho}_{r_p}^{\hat{\pi}}(s) - \rho_{r_p}^{\hat{\pi}}(s)\right| \le \frac{\varepsilon}{5}. \tag{12}$$

We control such concentration terms for both the constraint and reward value functions in Appendix C, and bound the terms in Eq. (12). In particular, we prove that for a fixed $\varepsilon \in (0, 1/1-\gamma]$, using $N \ge \tilde{O}\left(\frac{SA(B+H)}{\varepsilon^2\zeta^2}\right)$ ensures that the statements in Eq. (12) hold with probability $1 - 4\delta$. This guarantees that $\rho_r^{\pi^*}(s) - \rho_r^{\hat{\pi}}(s) \le \varepsilon$ and $\rho_c^{\hat{\pi}}(s) \ge b$. $\qquad\square$

## 6 LOWER-BOUND FOR WEAKLY COMMUNICATING CAMDPS

**Theorem 4** (Lower-bound for communicating CAMDP). For any sufficiently small $\varepsilon$, $\delta$, any sufficiently large $S$, $A$, and any $D \ge \max\{c_1 S, c_2\}$ (where $c_1, c_2 \ge 0$ is some universal constant), for any algorithm promising to return an $\frac{\varepsilon}{24}$-optimal policy with probability at least $\frac{3}{4}$ on any communicating CAMDP problem, there is a CAMDP such that the expected total samples on all state-action pairs, when running this algorithm, is at least $\tilde{\Omega}\left(\frac{SAH}{\varepsilon^2\zeta^2}\right)$

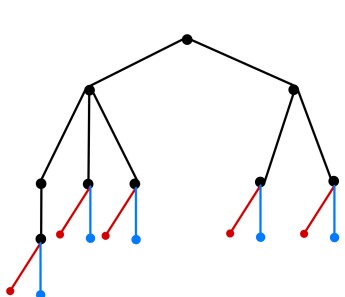
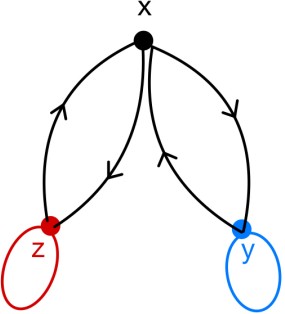

Figure 1: A Hard Communicating CAMDP when $A = 4$, $S = 19$.

Figure 2: A Component Communicating CAMDP.

*Proof Sketch:* We construct a family of hard CAMDP instances with parameters $S$, $A$, and diameter $D$. Define $A' := A - 1$, $D' := D/8$, and $K := \lceil S/4 \rceil$, and assume standard bounds: $A \ge 3$, $\varepsilon \le 1/16$, $D \ge \max\{16\lceil\log_A S\rceil, 16\}$.

We first design a primitive component MDP with three states $(x, y, z)$, each having $A'$ actions partitioned into subsets according to transition and reward structure (Figure 2). These components are embedded at the leaves of an $A'$-ary tree with $S - 3K$ internal nodes and depth at most $\lceil\log_{A'} S\rceil + 1$. The full MDP $M_0$ (Figure 1) connects components via deterministic transitions with diameter bounded by $D$. A collection of instances $\{M_{k,l}\}$ is constructed by perturbing action rewards at selected $x_k$ states. Optimal policies must distinguish between actions $a_1$ and $a_l$ at these states to satisfy the constraint. The divergence in occupancy measures under different instances implies a statistical gap. This separation in policy behavior across instances will be used to derive a lower bound. This separation arises from the amplification effect of the constraint reward $c$, which is necessary to ensure feasibility with respect to the objective defined in Eq. (1).

Finally, applying Fano's method Wainwright (2019) yields a minimax lower bound of $\tilde{\Omega}\left(\frac{SAD}{\varepsilon^2\zeta^2}\right)$, which translates to $\tilde{\Omega}\left(\frac{SAH}{\varepsilon^2\zeta^2}\right)$ under the bound $H \le D$ (Bartlett & Tewari, 2009). See Appendix G for a full proof. $\qquad\square$

## 7 LOWER-BOUND FOR GENERAL CAMDPS

**Theorem 5** (Lower-bound for general CAMDP). *For any sufficiently small $\varepsilon$, $\delta$, any sufficiently large $S$, $A$, for any algorithm promising to return an $\frac{\varepsilon}{24}$-optimal policy with probability at least $\frac{3}{4}$ on any communicating CAMDP problem, there is a CAMDP such that the expected total samples on all state-action pairs, when running this algorithm, is at least $\tilde{\Omega}\left(\frac{SA(H+B)}{\varepsilon^2\zeta^2}\right)$*

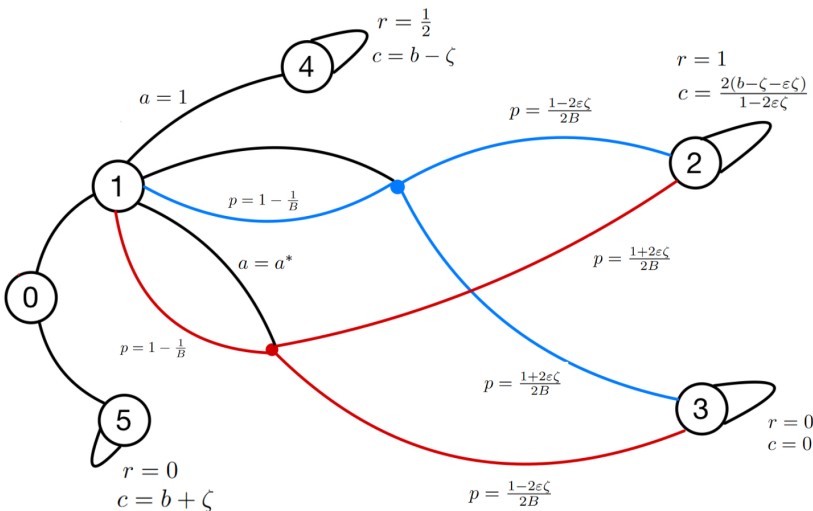

Figure 3: A Component MDP Used in the Hard Instance for CAMDP.

*Proof Sketch:* To establish the lower bound, we construct a family of hard instances in which achieving $\varepsilon/24$-average optimality requires significantly different policy behaviors across carefully designed environments. In particular, we show that a policy must choose action $a = 1$ in a designated subset of states with occupancy measure at most $2/3$ in one instance, while the same action must be selected with occupancy measure at least $2/3$ in another. This separation in policy behavior across instances will be used to derive a lower bound. This separation arises from the amplification effect of the constraint reward $c$, which is necessary to ensure feasibility with respect to the objective defined in Eq. (1). The design of our hard instance is motivated by the construction used for average-reward MDPs in Zurek & Chen (2024). Finally, applying Fano's inequality Wainwright (2019) to these instances yields a lower bound on the sample complexity of $\tilde{\Omega}\left(\frac{SAB}{\varepsilon^2\zeta^2}\right)$. Finally, by combining this result with Theorem 4, we obtain the general lower bound for weakly communicating CAMDPs: $\tilde{\Omega}\left(\frac{SA(B+H)}{\varepsilon^2\zeta^2}\right)$. See Appendix F for a full proof. □

## 8 CONCLUSION

In conclusion, we establish the **first minimax-optimal sample complexity bounds** for learning in CAMDPs under a generative model. Our algorithm operates under both relaxed and strict feasibility regimes, achieving tight upper bounds of $\tilde{O}\left(\frac{SA(B+H)}{\varepsilon^2}\right)$ and $\tilde{O}\left(\frac{SA(B+H)}{\varepsilon^2\zeta^2}\right)$, respectively. Complementing these results, we derive a matching lower bound of $\tilde{\Omega}\left(\frac{SA(B+H)}{\varepsilon^2\zeta^2}\right)$ for the strict feasibility setting, together with a specialized lower bound of $\tilde{\Omega}\left(\frac{SAH}{\varepsilon^2\zeta^2}\right)$ for the class of weakly communicating CAMDPs. Taken together, these results constitute the **first alignment of upper and lower bounds in all key problem parameters** — namely, the span bound of the bias function $H$, the transient time bound $B$, and the target accuracy $\varepsilon$. Our analysis therefore not only resolves the minimax sample complexity of CAMDPs for the first time, but also sheds new light on the fundamental complexity of constrained average-reward reinforcement learning, tightly connecting it to the structural properties of average-reward MDPs.

## 9 ACKNOWLEDGEMENTS

The research of Xudong Li was supported in part by the National Key R&D Program of China [Grant 2023YFA1009300] and the National Natural Science Foundation of China [Grants 12271107 and 12531014].

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

## A Proofs for primal-dual algorithm

**Theorem 1** (Guarantees for the primal-dual algorithm). *For a target error $\varepsilon_{\text{opt}} > 0$, consider the primal-dual updates given in Eq. (5)–Eq. (6) with parameters $U > |\lambda^*|$, $T = \frac{U^2}{\varepsilon_{\text{opt}}^2}\left[1 + \frac{1}{(U-\lambda^*)^2}\right]$, $\varepsilon_1 = \frac{\varepsilon_{\text{opt}}^2 (U-\lambda^*)}{6U}$ and $\eta = \frac{U}{\sqrt{T}}$, then the resulting mixture policy $\hat{\pi} := \frac{1}{T}\sum_{t=0}^{T-1}\hat{\pi}_t$ satisfies*

$$\rho_{r_p}^{\hat{\pi}}(s) \geq \rho_{r_p}^{\hat{\pi}^*}(s) - \varepsilon_{\text{opt}} \quad \text{and} \quad \rho_c^{\hat{\pi}}(s) \geq b' - \varepsilon_{\text{opt}}.$$

*Proof.* We will define the dual regret w.r.t $\lambda$ as the following quantity:

$$R^d(\lambda, T) := \sum_{t=0}^{T-1} (\lambda_t - \lambda)\left(\rho_c^{\hat{\pi}_t}(s) - b'\right). \tag{13}$$

Using the primal update in Eq. (5), for any $\pi$,

$$\rho_{r_p}^{\hat{\pi}_t}(s) + \lambda_t \rho_c^{\hat{\pi}_t}(s) \geq \rho_{r_p}^{\pi}(s) + \lambda_t \rho_c^{\pi}(s).$$

Substituting $\pi = \hat{\pi}^*$, we have,

$$\rho_{r_p}^{\hat{\pi}^*}(s) - \rho_{r_p}^{\hat{\pi}_t}(s) \leq \lambda_t \left[\rho_c^{\hat{\pi}_t}(s) - \rho_c^{\hat{\pi}^*}(s)\right].$$

Since $\hat{\pi}^*$ is a solution to the CAMDP, $\rho_c^{\hat{\pi}^*} \geq b'$, we get

$$\rho_{r_p}^{\hat{\pi}^*}(s) - \rho_{r_p}^{\hat{\pi}_t}(s) \leq \lambda_t \left[\rho_c^{\hat{\pi}_t}(s) - b'\right]. \tag{14}$$

Starting from the definition of the dual regret in Eq. (13), using Eq. (14) and dividing by $T$ gives

$$\frac{1}{T}\sum_{t=0}^{T-1}\left[\rho_{r_p}^{\hat{\pi}^*}(s) - \rho_{r_p}^{\hat{\pi}_t}(s)\right] + \frac{\lambda}{T}\sum_{t=0}^{T-1}\left(b' - \rho_c^{\hat{\pi}_t}(s)\right) \leq \frac{R^d(\lambda, T)}{T}. \tag{15}$$

Recall that $\hat{\pi} = \frac{1}{T}\sum_{t=0}^{T-1}\hat{\pi}_t$. Then, by the definition of this 'mixture', we have $\frac{1}{T}\sum_{t=0}^{T-1}\rho_{r_p}^{\hat{\pi}_t}(s) = \rho_{r_p}^{\hat{\pi}}(s)$ and $\frac{1}{T}\sum_{t=0}^{T-1}\rho_c^{\hat{\pi}_t}(s) = \rho_c^{\hat{\pi}}(s)$. Combining this with the last inequality, we get

$$\left[\rho_{r_p}^{\hat{\pi}^*}(s) - \rho_{r_p}^{\hat{\pi}}(s)\right] + \lambda\left(b' - \rho_c^{\hat{\pi}}(s)\right) \leq \frac{R^d(\lambda, T)}{T}. \tag{16}$$

Lemma 7 show that the following inequality holds for any $\lambda \in [0, U]$:

$$R^d(\lambda, T) \leq T^{3/2}\frac{\varepsilon_1^2 + 2\varepsilon_1 U}{2U} + U\sqrt{T}. \tag{17}$$

This combined with the previous inequality (and the "right" choice of $T$, the number of updates) gives the desired bounds. In particular, for the reward optimality gap, since $\lambda = 0 \in [0, U]$,

$$\rho_{r_p}^{\hat{\pi}^*}(s) - \rho_{r_p}^{\hat{\pi}}(s) \leq \sqrt{T}\frac{\varepsilon_1^2 + 2\varepsilon_1 U}{2U} + \frac{U}{\sqrt{T}} < \sqrt{T}\frac{3\varepsilon_1}{2} + \frac{U}{\sqrt{T}}. \qquad \text{(since } \varepsilon_1 < U\text{)}$$

For the constraint violation, there are two cases. The first case is when $b' - \rho_c^{\hat{\pi}}(s) \leq 0$. In this case, it also holds that $b' - \varepsilon_{\text{opt}} - \rho_c^{\hat{\pi}}(s) \leq 0$, which is what we wanted to show. The second case is when $b' - \rho_c^{\hat{\pi}}(\rho) > 0$. In this case, using the notation $[x]_+ = \max\{x, 0\}$ and Lemma 6, we have

$$\left[\rho_{r_p}^{\hat{\pi}^*}(s) - \rho_{r_p}^{\hat{\pi}}(s)\right] + U\left[b' - \rho_c^{\hat{\pi}}(s)\right]_+ \leq \frac{R^d(U, T)}{T}. \tag{18}$$

Because by assumption it holds that $U > \lambda^*$, Lemma 8 is applicable and gives that

$$\left[b' - \rho_c^{\hat{\pi}}(s)\right]_+ \leq \frac{R^d(U,T)}{T(U - \lambda^*)}. \tag{19}$$

Hence, since $U \in [0, U]$, combining the above display with Eq. (19) gives

$$\left[b' - \rho_c^{\hat{\pi}}(s)\right] \leq \left[b' - \rho_c^{\hat{\pi}}(s)\right]_+ \leq \sqrt{T}\,\frac{\varepsilon_1^2 + 2\varepsilon_1 U}{2U\,(U - \lambda^*)} + \frac{U}{(U - \lambda^*)\,\sqrt{T}} \tag{20}$$

$$< \sqrt{T}\,\frac{3\varepsilon_1}{2\,(U - \lambda^*)} + \frac{U}{(U - \lambda^*)\,\sqrt{T}}\,. \qquad \text{(since } \varepsilon_1 < U\text{)}$$

Now, set $T$ such that the second term in both quantities is bounded from above by $\varepsilon_{\mathrm{opt}}/2$. This gives

$$T = T_0 := \frac{U^2}{\varepsilon_{\mathrm{opt}}^2}\left[1 + \frac{1}{(U - \lambda^*)^2}\right]. \tag{21}$$

Now, set $\varepsilon_1$ such that the first term in both quantities is also bounded from above by $\frac{\varepsilon_{\mathrm{opt}}}{2}$. For this, choose

$$\varepsilon_1 = \frac{\varepsilon_{\mathrm{opt}}^2\,(U - \lambda^*)}{6U}\,.$$

With these values, the algorithm ensures that

$$\rho_{r_p}^{\hat{\pi}^*}(s) - \rho_{r_p}^{\hat{\pi}}(s) \leq \varepsilon_{\mathrm{opt}} \quad \text{and} \quad b' - \rho_c^{\hat{\pi}}(s) \leq \varepsilon_{\mathrm{opt}}. \tag{22}$$

$\square$

To further ensure the success of our primal-dual algorithm, we need to make sure $\lambda$ is bounded. So we obtains Lemma 6 as follows.

---

**Lemma 6** (Bounding the dual variable). *The objective Eq. (4) satisfies strong duality. Defining $\pi_c^* := \arg\max \rho_c^\pi(s)$. We consider two cases: (1) If $b' = b - \varepsilon'$ for $\varepsilon' > 0$ and event $\mathcal{E}_1 = \left\{\left|\hat{\rho}_c^{\pi_c^*} - \rho_c^{\pi_c^*}(s)\right| \leq \frac{\varepsilon'}{2}\right\}$ holds, then $\lambda^* \leq \frac{2(1+\omega)}{\varepsilon'}$ and (2) If $b' = b + \Delta$ for $\Delta \in \left(0, \frac{\zeta}{2}\right)$ and event $\mathcal{E}_2 = \left\{\left|\hat{\rho}_c^{\pi_c^*} - \rho_c^{\pi_c^*}(s)\right| \leq \frac{\zeta}{2} - \Delta\right\}$ holds, then $\lambda^* \leq \frac{2(1+\omega)}{\zeta}$.*

---

*Proof.* Writing the empirical CAMDP in Eq. (4) in its Lagrangian form,

$$\hat{\rho}_{r_p}^{\hat{\pi}^*}(s) = \max_\pi \min_{\lambda \geq 0} \hat{\rho}_{r_p}^\pi(s) + \lambda[\hat{\rho}_c^\pi(s) - b']$$

Using the linear programming formulation of CMDPs in terms of the state-occupancy measures $\mu$, we know that both the objective and the constraint are linear functions of $\mu$, and strong duality holds w.r.t $\mu$. Since $\mu$ and $\pi$ have a one-one mapping, we can switch the min and the max (Paternain et al., 2019), implying,

$$= \min_{\lambda \geq 0} \max_\pi \hat{\rho}_{r_p}^\pi(s) + \lambda[\hat{\rho}_c^\pi(s) - b']$$

Since $\lambda^*$ is the optimal dual variable for the empirical CMDP in Eq. (4),

$$= \max_\pi \hat{\rho}_{r_p}^\pi(s) + \lambda^*\left[\hat{\rho}_c^\pi(s) - b'\right]$$

Define $\pi_c^* := \arg\max \rho_c^\pi(s)$ and $\hat{\pi}_c^* := \arg\max \hat{\rho}_c^\pi(s)$

$$\geq \hat{\rho}_{r_p}^{\hat{\pi}_c^*}(s) + \lambda^*\left[\hat{\rho}_c^{\hat{\pi}_c^*}(s) - b'\right]$$

$$= \hat{\rho}_{r_p}^{\hat{\pi}_c^*}(s) + \lambda^*\left[\left(\hat{\rho}_c^{\hat{\pi}_c^*}(s) - \rho_c^{\pi_c^*}(s)\right) + (\rho_c^{\pi_c^*}(s) - b) + (b - b')\right]$$

By definition, $\zeta = \rho_c^{\pi_c^*}(s) - b$

$$= \hat{\rho}_{r_p}^{\hat{\pi}_c^*}(s) + \lambda^*\left[\left(\hat{\rho}_c^{\hat{\pi}_c^*}(s) - \hat{\rho}_c^{\pi_c^*}(s)\right) + \left(\hat{\rho}_c^{\pi_c^*}(s) - \rho_c^{\pi_c^*}(s)\right) + \zeta + (b - b')\right]$$

By definition of $\hat{\pi}_c^*$, $\left(\hat{\rho}_c^{\hat{\pi}_c^*}(s) - \hat{\rho}_c^{\pi_c^*}(s)\right) \geq 0$

$$\hat{\rho}_{r_p}^{\hat{\pi}^*}(s) \geq \hat{\rho}_{r_p}^{\hat{\pi}_c^*}(s) + \lambda^* \left[\zeta + (b - b') - \left|\hat{\rho}_c^{\pi_c^*}(s) - \rho_c^{\pi_c^*}(s)\right|\right]$$

1) If $b' = b - \varepsilon'$ for $\varepsilon' > 0$. Hence,

$$\hat{\rho}_{r_p}^{\hat{\pi}^*}(s) \geq \hat{\rho}_{r_p}^{\hat{\pi}_c^*}(s) + \lambda^* \left[\zeta + \varepsilon' - \left|\hat{\rho}_c^{\pi_c^*}(s) - \rho_c^{\pi_c^*}(s)\right|\right]$$

If the event $\mathcal{E}_1$ holds, $\left|\hat{\rho}_c^{\pi_c^*}(s) - \rho_c^{\pi_c^*}(s)\right| \leq \frac{\varepsilon'}{2}$, implying, $\left|\hat{\rho}_c^{\pi_c^*}(s) - \rho_c^{\pi_c^*}(s)\right| < \zeta + \frac{\varepsilon'}{2}$, then,

$$\geq \hat{\rho}_{r_p}^{\hat{\pi}_c^*}(s) + \lambda^* \frac{\varepsilon'}{2}$$

$$\implies \lambda^* \leq \frac{2}{\varepsilon'}[\hat{\rho}_{r_p}^{\hat{\pi}^*}(s) - \hat{\rho}_{r_p}^{\hat{\pi}_c^*}(s)] \leq \frac{2(1+\omega)}{\varepsilon'}$$

2) If $b' = b + \Delta$ for $\Delta \in \left(0, \frac{\zeta}{2}\right)$. Hence,

$$\hat{\rho}_{r_p}^{\hat{\pi}^*}(s) \geq \hat{\rho}_{r_p}^{\hat{\pi}_c^*}(s) + \lambda^* \left[\zeta - \Delta - \left|\hat{\rho}_c^{\pi_c^*}(s) - \rho_c^{\pi_c^*}(s)\right|\right]$$

If the event $\mathcal{E}_2$ holds, $\left|\hat{\rho}_c^{\pi_c^*}(s) - \rho_c^{\pi_c^*}(s)\right| \leq \frac{\zeta}{2} - \Delta$ for $\Delta < \frac{\zeta}{2}$, then,

$$\geq \hat{\rho}_{r_p}^{\hat{\pi}_c^*}(s) + \lambda^* \frac{\zeta}{2}$$

$$\implies \lambda^* \leq \frac{2}{\zeta}[\hat{\rho}_{r_p}^{\hat{\pi}^*}(s) - \hat{\rho}_{r_p}^{\hat{\pi}_c^*}(s)] \leq \frac{2(1+\omega)}{\zeta}$$

---

**Lemma 7** (Bounding the dual regret). *For the dual regret defined in Eq.* (13)*, we have*
$$R^d(\lambda, T) \leq T^{3/2} \frac{\varepsilon_l^2 + 2\varepsilon_l U}{2U} + U\sqrt{T}.$$

---

*Proof.* First, fix an arbitrary $\lambda \in [0, U]$. Defining $\lambda'_{t+1} := \mathbb{P}_{[0,U]}[\lambda_t - \eta(\hat{\rho}_c^{\hat{\pi}_t}(s) - b')]$,

So we have,
$$|\lambda_{t+1} - \lambda| = |\mathcal{R}_\Lambda[\lambda'_{t+1}] - \lambda| = |\mathcal{R}_\Lambda[\lambda'_{t+1}] - \lambda'_{t+1} + \lambda'_{t+1} - \lambda| \leq |\mathcal{R}_\Lambda[\lambda'_{t+1}] - \lambda'_{t+1}| + |\lambda'_{t+1} - \lambda|$$
$$\leq \varepsilon_l + |\lambda'_{t+1} - \lambda|.$$
$$\text{(since } |\lambda - \mathcal{R}_\Lambda[\lambda]| \leq \varepsilon_l \text{ for all } \lambda \in [0, U] \text{ because of the epsilon-net.)}$$

Squaring both sides,
$$|\lambda_{t+1} - \lambda|^2 = \varepsilon_l^2 + |\lambda'_{t+1} - \lambda|^2 + 2\varepsilon_l |\lambda'_{t+1} - \lambda| \leq \varepsilon_l^2 + 2\varepsilon_l U + |\lambda'_{t+1} - \lambda|^2$$
$$\text{(since } \lambda, \lambda'_{t+1} \in [0, U]\text{,)}$$
$$\leq \varepsilon_l^2 + 2\varepsilon_l U + |\lambda_t - \eta(\hat{\rho}_c^{\hat{\pi}_t}(s) - b') - \lambda|^2 \quad \text{(since projections are non-expansive)}$$
$$= \varepsilon_l^2 + 2\varepsilon_l U + |\lambda_t - \lambda|^2 - 2\eta(\lambda_t - \lambda)(\hat{\rho}_c^{\hat{\pi}_t}(s) - b') + \eta^2(\hat{\rho}_c^{\hat{\pi}_t}(s) - b')^2$$
$$\leq \varepsilon_l^2 + 2\varepsilon_l U + |\lambda_t - \lambda|^2 - 2\eta(\lambda_t - \lambda)(\hat{\rho}_c^{\hat{\pi}_t}(s) - b') + \eta^2,$$
where the last inequality follows because $b'$ and the constraint value are in the $[0, 1]$ interval. Rearranging and dividing by $2\eta$, we get
$$(\lambda_t - \lambda)(\hat{\rho}_c^{\hat{\pi}_t}(s) - b') \leq \frac{\varepsilon_l^2 + 2\varepsilon_l U}{2\eta} + \frac{|\lambda_t - \lambda|^2 - |\lambda_{t+1} - \lambda|^2}{2\eta} + \frac{\eta}{2}.$$
Summing from $t = 0$ to $T - 1$ and using the definition of the dual regret,
$$R^d(\lambda, T) \leq T \frac{\varepsilon_l^2 + 2\varepsilon_l U}{2\eta} + \frac{1}{2\eta} \sum_{t=0}^{T-1} [|\lambda_t - \lambda|^2 - |\lambda_{t+1} - \lambda|^2] + \frac{\eta T}{2}.$$
Telescoping, bounding $|\lambda_0 - \lambda|$ by $U$ and dropping a negative term gives
$$R^d(\lambda, T) \leq T \frac{\varepsilon_l^2 + 2\varepsilon_l U}{2\eta} + \frac{U^2}{2\eta} + \frac{\eta T}{2}.$$
Setting $\eta = \frac{U}{\sqrt{T}}$,
$$R^d(\lambda, T) \leq T^{3/2} \frac{\varepsilon_l^2 + 2\varepsilon_l U}{2U} + U\sqrt{T}, \tag{23}$$
which finishes the proof. □

**Lemma 8** (Bounding the positive constraint value). *For any $C > \lambda^*$ and any $\tilde{\pi}$ s.t. $\rho_r^{\hat{\pi}^*}(s) - \rho_r^{\tilde{\pi}}(s) + C[b' - \varepsilon_{opt} - \rho_c^{\tilde{\pi}}(s)]_+ \leq \beta$, we have $[b' - \varepsilon_{opt} - \rho_c^{\tilde{\pi}}(s)]_+ \leq \frac{\beta}{C - \lambda^*}$.*

*Proof.* Define $\nu(\tau) = \max_{\pi} \{\rho_r^{\pi}(s) \mid \rho_c^{\pi}(s) \geq b' - \varepsilon_{\text{opt}} + \tau\}$ and note that by definition, $\nu(0) = \rho_r^{\tilde{\pi}^*}(s)$ and that $\nu$ is a decreasing function for its argument.

Let $\rho_l^{\pi,\lambda}(s) = \rho_r^{\pi}(s) + \lambda(\rho_c^{\pi}(s) - b' - \varepsilon_{\text{opt}})$. Then, for any policy $\pi$ s.t. $\rho_c^{\pi}(s) \geq b' - \varepsilon_{\text{opt}} + \tau$, we have

$$\rho_l^{\pi,\lambda^*}(s) \leq \max_{\pi'} \rho_l^{\pi',\lambda^*}(s)$$

$$= \rho_r^{\tilde{\pi}^*}(s) \qquad \text{(by strong duality)}$$

$$= \nu(0) \qquad \text{(from above relation)}$$

$$\implies \nu(0) - \tau\lambda^* \geq \rho_l^{\pi,\lambda}(s) - \tau\lambda^* = \rho_r^{\pi}(s) + \lambda^* \underbrace{(\rho_c^{\pi}(s) - b' + \varepsilon_{\text{opt}} - \tau)}_{\text{Non-negative}}$$

$$\implies \nu(0) - \tau\lambda^* \geq \max_{\pi} \{\rho_r^{\pi}(s) \mid \rho_c^{\pi}(s) \geq b' - \varepsilon_{\text{opt}} + \tau\} = \nu(\tau).$$

$$\implies \tau\lambda^* \leq \nu(0) - \nu(\tau). \tag{24}$$

Now we choose $\tilde{\tau} = -(b' - \varepsilon_{\text{opt}} - \rho_c^{\tilde{\pi}}(s))_+$.

$$(C - \lambda^*)|\tilde{\tau}| = \lambda^*\tilde{\tau} + C|\tilde{\tau}| \qquad \text{(since } \tilde{\tau} \leq 0\text{)}$$

$$\leq \nu(0) - \nu(\tilde{\tau}) + C|\tilde{\tau}| \qquad \text{(Eq. (24))}$$

$$= \rho_r^{\tilde{\pi}^*}(s) - \rho_r^{\tilde{\pi}}(s) + C|\tilde{\tau}| + \rho_r^{\tilde{\pi}}(s) - \nu(\tilde{\tau}) \qquad \text{(definition of } \nu(0)\text{)}$$

$$= \rho_r^{\tilde{\pi}^*}(s) - \rho_r^{\tilde{\pi}}(s) + C(b' - \varepsilon_{\text{opt}} - \rho_c^{\tilde{\pi}}(s))_+ + \rho_r^{\tilde{\pi}}(s) - \nu(\tilde{\tau})$$

$$\leq \beta + \rho_r^{\tilde{\pi}}(s) - \nu(\tilde{\tau}).$$

Now let us bound $\nu(\tilde{\tau})$:

$$\nu(\tilde{\tau}) = \max_{\pi} \{\rho_r^{\pi}(s) \mid \rho_c^{\pi}(s) \geq b' - \varepsilon_{\text{opt}} - (b' - \varepsilon_{\text{opt}} - \rho_c^{\tilde{\pi}}(s))_+\}$$

$$\geq \max_{\pi} \{\rho_r^{\pi}(s) \mid \rho_c^{\pi}(s) \geq \rho_c^{\tilde{\pi}}(s)\} \qquad \text{(tightening the constraint)}$$

$$\nu(\tilde{\tau}) \geq \rho_r^{\tilde{\pi}}(s) \implies (C - \lambda^*)|\tilde{\tau}| \leq \beta \implies (b' - \varepsilon_{\text{opt}} - \rho_c^{\tilde{\pi}}(s))_+ \leq \frac{\beta}{C - \lambda^*}$$

$\square$

# B   PROOF OF THEOREM 2

**Theorem 2.** *For a fixed $\varepsilon \in (0, 1]$, $\delta \in (0, 1)$ and a general CAMDP, suppose the corresponding AMDPs $(\mathcal{P}, r)$ and $(\mathcal{P}, c)$ have bias functions bound $H$ , and satisfy the bounded transient time assumption with parameter $B$. Algorithm 1 with $N = \tilde{O}\left(\frac{SA(B+H)}{\varepsilon^2}\right)$ samples, $b' = b - \frac{3\varepsilon}{8}$, $\omega = \frac{\varepsilon(1-\gamma)}{8}$, $U = O\left(\frac{1}{\varepsilon}(1-\gamma)\right)$, $\varepsilon_1 = O\left(\varepsilon^2(1-\gamma)^2\right)$, $T = O\left(\frac{1}{(1-\gamma)^4\varepsilon^4}\right)$ and $\gamma = 1 - \frac{\varepsilon_{\text{opt}}}{4(B+H)}$, returns policy $\hat{\pi}$ that satisfies the objective in Eq. (2) with probability at least $1 - 4\delta$.*

*Proof.* We fill in the details required for the proof sketch in the main paper. Proceeding according to the proof sketch, we first detail the computation of $T$ and $\varepsilon_1$ for the primal-dual algorithm. Recall that $U = \frac{32}{5\varepsilon(1-\gamma)}$ and $\varepsilon_{\text{opt}} = \frac{\varepsilon}{4}$. Using Theorem 1, we need to set

$$T = \frac{4U^2}{\varepsilon_{\text{opt}}^2(1-\gamma)^2}\left[1 + \frac{1}{(U-\lambda^*)^2}\right] = \frac{64}{\varepsilon^2(1-\gamma)^2}\left[1 + \frac{1}{(U-\lambda^*)^2}\right]$$

Recall that $|\lambda^*| \leq C := \frac{16}{5\varepsilon(1-\gamma)}$ and $U = 2C$. Simplifying,

$$\leq \frac{256}{\varepsilon^2(1-\gamma)^2}\left[C^2 + 1\right] < \frac{512}{\varepsilon^2(1-\gamma)^2}C^2 = \frac{512}{\varepsilon^2(1-\gamma)^2}\frac{256}{25\varepsilon^2(1-\gamma)^2}$$

$$\implies T = O\left(\frac{1}{\varepsilon^4(1-\gamma)^4}\right).$$

Using Theorem 1, we need to set $\varepsilon_1$,

$$\varepsilon_1 = \frac{\varepsilon_{\text{opt}}^2(1-\gamma)^2(U-\lambda^*)}{6U} = \frac{\varepsilon^2(1-\gamma)^2(U-\lambda^*)}{96U} \leq \frac{\varepsilon^2(1-\gamma)^2}{96}$$

$$\implies \varepsilon_1 = O\left(\varepsilon^2(1-\gamma)^2\right).$$

For bounding the concentration terms for $\hat{\pi}$ in Eq. (9), we first use Lemma 11 to convert them to discounted setting, then use Lemma 13 with $U = \frac{32}{5\varepsilon(1-\gamma)}$, $\omega = \frac{\varepsilon(1-\gamma)}{8}$ and $\varepsilon_1 = \frac{\varepsilon^2(1-\gamma)^2}{96}$. In this case, $\iota = \frac{\omega\,\delta\,(1-\gamma)\,\varepsilon_1}{30\,U|S||A|^2} = O\left(\frac{\delta\varepsilon^4\,(1-\gamma)^4}{SA^2}\right)$ and in order to satisfy the concentration bounds for $\hat{\pi}$, we require that

$$N \geq \tilde{O}\left(\frac{SA(B+H)}{\varepsilon^2}\right)$$

We use the Lemma 14 to bound the remaining concentration terms for $\pi^*$ and $\pi_c^*$ in Eq. (9). In this case, for $C'(\delta) = 72\log\left(\frac{4S\log(e/1-\gamma)}{\delta}\right)$, we require that,

$$N \geq \tilde{O}\left(\frac{SA(B+H)}{\varepsilon^2}\right)$$

Hence, if $N \geq \tilde{O}\left(\frac{SA(B+H)}{\varepsilon^2}\right)$, the bounds in Eq. (9) are satisfied, completing the proof. □

---

**Lemma 9** (Decomposing the suboptimality). *For $b' = b - \frac{\varepsilon-\varepsilon_{opt}}{2}$, if (i) $\varepsilon_{opt} < \varepsilon$, and (ii) the following conditions are satisfied,*

$$\left|\rho_c^{\hat{\pi}}(s) - \hat{\rho}_c^{\hat{\pi}}(s)\right| \leq \frac{\varepsilon - \varepsilon_{opt}}{2}\; ; \; \left|\rho_c^{\pi^*}(s) - \hat{\rho}_c^{\pi^*}(s)\right| \leq \frac{\varepsilon - \varepsilon_{opt}}{2}$$

*where $\pi_c^* := \arg\max \rho_c^\pi(s)$, then (a) policy $\hat{\pi}$ violates the constraint by at most $\varepsilon$ i.e. $\rho_c^{\hat{\pi}}(s) \geq b - \varepsilon$ and (b) its optimality gap can be bounded as:*

$$\rho_r^{\pi^*}(s) - \rho_r^{\hat{\pi}}(s) \leq 2\omega + \varepsilon_{opt} + \left|\rho_{r_p}^{\pi^*}(s) - \hat{\rho}_{r_p}^{\pi^*}(s)\right| + \left|\hat{\rho}_{r_p}^{\hat{\pi}}(s) - \rho_{r_p}^{\hat{\pi}}(s)\right|$$

*Proof.* From Theorem 1, we know that,

$$\hat{\rho}_c^{\hat{\pi}}(s) \geq b' - \varepsilon_{\text{opt}} \implies \rho_c^{\hat{\pi}}(s) \geq \rho_c^{\hat{\pi}}(s) - \hat{\rho}_c^{\hat{\pi}}(s) + b' - \varepsilon_{\text{opt}} \geq -\left|\rho_c^{\hat{\pi}}(s) - \hat{\rho}_c^{\hat{\pi}}(s)\right| + b' - \varepsilon_{\text{opt}}$$

Since we require $\hat{\pi}$ to violate the constraint in the true CMDP by at most $\varepsilon$, we require $\rho_c^{\hat{\pi}}(s) \geq b - \varepsilon$. From the above equation, a sufficient condition for ensuring this is,

$$- \left|\rho_c^{\hat{\pi}}(s) - \hat{\rho}_c^{\hat{\pi}}(s)\right| + b' - \varepsilon_{\text{opt}} \geq b - \varepsilon \,,$$

meaning that we require

$$\left|\rho_c^{\hat{\pi}}(s) - \hat{\rho}_c^{\hat{\pi}}(s)\right| \leq (b' - b) - \varepsilon_{\text{opt}} + \varepsilon.$$

Plugging in the value of $b'$, we see that this sufficient condition indeed holds, by our assumption that $\left|\rho_c^{\hat{\pi}}(s) - \hat{\rho}_c^{\hat{\pi}}(s)\right| \leq \frac{\varepsilon - \varepsilon_{\text{opt}}}{2}$.

Let $\pi^*$ be the solution to Eq. (1). Our next goal is to show that $\pi^*$ is feasible for the constrained problem in Eq. (4), i.e., $\hat{\rho}_c^{\pi^*}(s) \geq b'$. We have

$$\rho_c^{\pi^*}(s) \geq b \implies \hat{\rho}_c^{\pi^*}(s) \geq b - \left|\rho_c^{\pi^*}(s) - \hat{\rho}_c^{\pi^*}(s)\right|$$

Since we require $\hat{\rho}_c^{\pi^*}(s) \geq b'$, using the above equation, a sufficient condition to ensure this is

$$b - \left|\rho_c^{\pi^*}(s) - \hat{\rho}_c^{\pi^*}(s)\right| \geq b' \text{meaning that we require } \left|\rho_c^{\pi^*}(s) - \hat{\rho}_c^{\pi^*}(s)\right| \leq b - b'.$$

Since $b' = b - \frac{\varepsilon - \varepsilon_{\text{opt}}}{2}$, we require that

$$\left|\rho_c^{\pi^*}(s) - \hat{\rho}_c^{\pi^*}(s)\right| \leq \frac{\varepsilon - \varepsilon_{\text{opt}}}{2}.$$

Given that the above statements hold, we can decompose the suboptimality in the reward value function as follows:

$$\rho_r^{\pi^*}(s) - \rho_r^{\hat{\pi}}(s)$$
$$= \rho_r^{\pi^*}(s) - \rho_{r_p}^{\pi^*}(s) + \rho_{r_p}^{\pi^*}(s) - \rho_r^{\hat{\pi}}(s)$$
$$= [\rho_r^{\pi^*}(s) - \rho_{r_p}^{\pi^*}(s)] + \rho_{r_p}^{\pi^*}(s) - \hat{\rho}_{r_p}^{\pi^*}(s) + \hat{\rho}_{r_p}^{\pi^*}(s) - \rho_r^{\hat{\pi}}(s)$$
$$\leq [\rho_r^{\pi^*}(s) - \rho_{r_p}^{\pi^*}(s)] + [\rho_{r_p}^{\pi^*}(s) - \hat{\rho}_{r_p}^{\pi^*}(s)] + \hat{\rho}_{r_p}^{\hat{\pi}^*}(s) - \rho_r^{\hat{\pi}}(s)$$
$$\text{(By optimality of } \hat{\pi}^* \text{ and since we have ensured that } \pi^* \text{ is feasible for Eq. (4))}$$
$$= [\rho_r^{\pi^*}(s) - \rho_{r_p}^{\pi^*}(s)] + [\rho_{r_p}^{\pi^*}(s) - \hat{\rho}_{r_p}^{\pi^*}(s)] + [\hat{\rho}_{r_p}^{\hat{\pi}^*}(s) - \hat{\rho}_{r_p}^{\hat{\pi}}(s)] + \hat{\rho}_{r_p}^{\hat{\pi}}(s) - \rho_r^{\hat{\pi}}(s)$$
$$= \underbrace{[\rho_r^{\pi^*}(s) - \rho_{r_p}^{\pi^*}(s)]}_{\text{Perturbation Error}} + \underbrace{[\rho_{r_p}^{\pi^*}(s) - \hat{\rho}_{r_p}^{\pi^*}(s)]}_{\text{Concentration Error}} + \underbrace{[\hat{\rho}_{r_p}^{\hat{\pi}^*}(s) - \hat{\rho}_{r_p}^{\hat{\pi}}(s)]}_{\text{Primal-Dual Error}} + \underbrace{[\hat{\rho}_{r_p}^{\hat{\pi}}(s) - \rho_{r_p}^{\hat{\pi}}(s)]}_{\text{Concentration Error}} + \underbrace{[\rho_{r_p}^{\hat{\pi}}(s) - \rho_r^{\hat{\pi}}(s)]}_{\text{Perturbation Error}}$$

For a perturbation magnitude equal to $\omega$, we can bound both perturbation errors by $\omega$. Using Theorem 1 to bound the primal-dual error by $\varepsilon_{\text{opt}}$,

$$\rho_r^{\pi^*}(s) - \rho_r^{\hat{\pi}}(s) \leq 2\omega + \varepsilon_{\text{opt}} + \underbrace{[\rho_{r_p}^{\pi^*}(s) - \hat{\rho}_{r_p}^{\pi^*}(s)]}_{\text{Concentration Error}} + \underbrace{[\hat{\rho}_{r_p}^{\hat{\pi}}(s) - \rho_{r_p}^{\hat{\pi}}(s)]}_{\text{Concentration Error}}.$$

$\square$

## C  PROOF OF THEOREM 3

**Theorem 3.** For a fixed $\varepsilon \in (0, 1/1-\gamma]$ and $\delta \in (0,1)$, Algorithm 1, with $N = \tilde{O}\left(\frac{SA(B+H)}{\varepsilon^2 \zeta^2}\right)$ samples, $b' = b + \frac{\varepsilon(1-\gamma)\zeta}{20}, \omega = \frac{\varepsilon(1-\gamma)}{10}, U = \frac{4(1+\omega)}{\zeta(1-\gamma)}, \varepsilon_1 = O\left(\varepsilon^2(1-\gamma)^4\zeta^2\right), T = O\left(1/(1-\gamma)^6\zeta^4\varepsilon^2\right)$ and $\gamma = 1 - \frac{\varepsilon_{opt}}{4(B+H)}$ returns policy $\hat{\pi}$ that satisfies the objective in Eq. (3), with probability at least $1 - 4\delta$.

*Proof.* We fill in the details required for the proof sketch in the main paper. Proceeding according to the proof sketch, we first detail the computation of $T$ and $\varepsilon_1$ for the primal-dual algorithm. Recall that $U = \frac{8}{\zeta(1-\gamma)}, \Delta = \frac{\varepsilon(1-\gamma)\zeta}{40}$ and $\varepsilon_{opt} = \frac{\Delta}{5}$. Using Theorem 1, we need to set

$$T = \frac{4U^2}{\varepsilon_{opt}^2(1-\gamma)^2}\left[1 + \frac{1}{(U-\lambda^*)^2}\right] = \frac{100}{\Delta^2(1-\gamma)^2}\left[1 + \frac{1}{(U-\lambda^*)^2}\right]$$

Recall that $|\lambda^*| \leq C := \frac{4}{\zeta(1-\gamma)}$ and $U = 2C$. Simplifying,

$$\leq \frac{400}{\Delta^2(1-\gamma)^2}\left[C^2 + 1\right] < \frac{800}{\Delta^2(1-\gamma)^2}C^2 = \frac{800}{\Delta^2(1-\gamma)^2}\frac{16}{\zeta^2(1-\gamma)^2}$$

$$\implies T \leq \frac{800 \cdot 1600}{\varepsilon^2\zeta^2(1-\gamma)^4}\frac{16}{\zeta^2(1-\gamma)^2} = O\left(1/\varepsilon^2\zeta^4(1-\gamma)^6\right).$$

Using Theorem 1, we need to set $\varepsilon_1$,

$$\varepsilon_1 = \frac{\varepsilon_{opt}^2(1-\gamma)^2(U-\lambda^*)}{6U} = \frac{\Delta^2(1-\gamma)^2(U-\lambda^*)}{150U} \leq \frac{\Delta^2(1-\gamma)^2}{150}$$

$$\implies \varepsilon_1 \leq \frac{\varepsilon^2\zeta^2(1-\gamma)^4}{150\cdot1600} = O\left(\varepsilon^2\zeta^2(1-\gamma)^4\right).$$

For bounding the concentration terms for $\hat{\pi}$ in Eq. (12), we first use Lemma 11 to convert them to discounted setting, then use Lemma 13 with $U = \frac{8}{\zeta(1-\gamma)}, \omega = \frac{\varepsilon(1-\gamma)}{10}$ and $\varepsilon_1 = \frac{\varepsilon^2\zeta^2(1-\gamma)^4}{150\cdot1600}$. In this case, $\iota = \frac{\omega\delta(1-\gamma)\varepsilon_1}{30U|S||A|^2} = O\left(\frac{\delta\varepsilon^3\zeta^3(1-\gamma)^7}{SA^2}\right)$ and in order to satisfy the concentration bounds for $\hat{\pi}$, we require that

$$\tilde{O}\left(\frac{SA(B+H)}{\varepsilon^2\zeta^2}\right)$$

We use the Lemma 14 to bound the remaining concentration terms for $\pi^*$ and $\pi_c^*$ in Eq. (12). In this case, for $C'(\delta) = 72\log\left(\frac{4S\log(e/1-\gamma)}{\delta}\right)$, we require that,

$$\tilde{O}\left(\frac{SA(B+H)}{\varepsilon^2\zeta^2}\right)$$

Hence, if $N \geq \tilde{O}\left(\frac{SA(B+H)}{\varepsilon^2\zeta^2}\right)$, the bounds in Eq. (12) are satisfied, completing the proof. $\square$

---

**Lemma 10** (Decomposing the suboptimality). *For a fixed $\Delta > 0$ and $\varepsilon_{opt} < \Delta$, if $b' = b + \Delta$, then the following conditions are satisfied,*

$$\left|\rho_c^{\hat{\pi}}(s) - \hat{\rho}_c^{\hat{\pi}}(s)\right| \leq \Delta - \varepsilon_{opt}; \left|\rho_c^{\pi^*}(s) - \hat{\rho}_c^{\pi^*}(s)\right| \leq \Delta$$

*then (a) policy $\hat{\pi}$ satisfies the constraint i.e. $\rho_c^{\hat{\pi}}(s) \geq b$ and (b) its optimality gap can be bounded as:*

$$\rho_r^{\pi^*}(s) - \rho_r^{\hat{\pi}}(s) \leq 2\omega + \varepsilon_{opt} + 2\Delta\lambda^* + \left|\rho_{r_p}^{\pi^*}(s) - \hat{\rho}_{r_p}^{\pi^*}(s)\right| + \left|\hat{\rho}_{r_p}^{\hat{\pi}}(s) - \rho_{r_p}^{\hat{\pi}}(s)\right|.$$

---

*Proof.* Compared to Eq. (4), we define a slightly modified CMDP problem by changing the constraint RHS to $b''$ for some $b''$ to be specified later. We denote its corresponding optimal policy as $\tilde{\pi}^*$. In particular,

$$\tilde{\pi}^* \in \arg\max_\pi \hat{\rho}_{r_p}^\pi(s) \text{ s.t. } \hat{\rho}_c^\pi(s) \geq b'' \tag{25}$$

From Theorem 1, we know that,

$$\hat{\rho}_c^{\hat{\pi}}(s) \geq b' - \varepsilon_{opt} \implies \rho_c^{\hat{\pi}}(s) \geq \rho_c^{\hat{\pi}}(s) - \hat{\rho}_c^{\hat{\pi}}(s) + b' - \varepsilon_{opt} \geq -\left|\rho_c^{\hat{\pi}}(s) - \hat{\rho}_c^{\hat{\pi}}(s)\right| + b' - \varepsilon_{opt}$$

Since we require $\hat{\pi}$ to satisfy the constraint in the true CMDP, we require $\rho_c^{\hat{\pi}}(s) \geq b$. From the above equation, a sufficient condition for ensuring this is,

$$- \left| \rho_c^{\hat{\pi}}(s) - \hat{\rho}_c^{\hat{\pi}}(s) \right| + b' - \varepsilon_{\text{opt}} \geq b$$

meaning that we require $\left| \rho_c^{\hat{\pi}}(s) - \hat{\rho}_c^{\hat{\pi}}(s) \right| \leq (b' - b) - \varepsilon_{\text{opt}}$.

In the subsequent analysis, we will require $\pi^*$ to be feasible for the constrained problem in Eq. (25). This implies that we require $\hat{\rho}_c^{\pi^*}(s) \geq b''$. Since $\pi^*$ is the solution to Eq. (1), we know that,

$$\rho_c^{\pi^*}(s) \geq b \implies \hat{\rho}_c^{\pi^*}(s) \geq b - \left| \rho_c^{\pi^*}(s) - \hat{\rho}_c^{\pi^*}(s) \right|$$

Since we require $\hat{\rho}_c^{\pi^*}(s) \geq b''$, using the above equation, a sufficient condition to ensure this is

$$b - \left| \rho_c^{\pi^*}(s) - \hat{\rho}_c^{\pi^*}(s) \right| \geq b'' \text{meaning that we require } \left| \rho_c^{\pi^*}(s) - \hat{\rho}_c^{\pi^*}(s) \right| \leq b - b''.$$

Hence we require the following statements to hold:

$$\left| \rho_c^{\hat{\pi}}(s) - \hat{\rho}_c^{\hat{\pi}}(s) \right| \leq (b' - b) - \varepsilon_{\text{opt}} \quad ; \quad \left| \rho_c^{\pi^*}(s) - \hat{\rho}_c^{\pi^*}(s) \right| \leq b - b''.$$

Given that the above statements hold, we can decompose the suboptimality in the reward value function as follows:

$$\rho_r^{\pi^*}(s) - \rho_r^{\hat{\pi}}(s) = \rho_r^{\pi^*}(s) - \rho_{r_p}^{\pi^*}(s) + \rho_{r_p}^{\pi^*}(s) - \rho_r^{\hat{\pi}}(s)$$

$$= [\rho_r^{\pi^*}(s) - \rho_{r_p}^{\pi^*}(s)] + [\rho_{r_p}^{\pi^*}(s) - \hat{\rho}_{r_p}^{\pi^*}(s)] + \hat{\rho}_{r_p}^{\pi^*}(s) - \rho_r^{\hat{\pi}}(s)$$

$$\leq [\rho_r^{\pi^*}(s) - \rho_{r_p}^{\pi^*}(s)] + [\rho_{r_p}^{\pi^*}(s) - \hat{\rho}_{r_p}^{\pi^*}(s)] + \hat{\rho}_{r_p}^{\hat{\pi}^*}(s) - \rho_r^{\hat{\pi}}(s)$$

(By optimality of $\hat{\pi}^*$ and since we have ensured that $\pi^*$ is feasible for Eq. (25))

$$= [\rho_r^{\pi^*}(s) - \rho_{r_p}^{\pi^*}(s)] + [\rho_{r_p}^{\pi^*}(s) - \hat{\rho}_{r_p}^{\pi^*}(s)] + [\hat{\rho}_{r_p}^{\tilde{\pi}^*}(s) - \hat{\rho}_{r_p}^{\hat{\pi}^*}(s)] + \hat{\rho}_{r_p}^{\hat{\pi}^*}(s) - \rho_r^{\hat{\pi}}(s)$$

$$= [\rho_r^{\pi^*}(s) - \rho_{r_p}^{\pi^*}(s)] + [\rho_{r_p}^{\pi^*}(s) - \hat{\rho}_{r_p}^{\pi^*}(s)] + [\hat{\rho}_{r_p}^{\tilde{\pi}^*}(s) - \hat{\rho}_{r_p}^{\hat{\pi}^*}(s)] + \hat{\rho}_{r_p}^{\hat{\pi}^*}(s) - \rho_r^{\hat{\pi}}(s)$$

$$= [\rho_r^{\pi^*}(s) - \rho_{r_p}^{\pi^*}(s)] + [\rho_{r_p}^{\pi^*}(s) - \hat{\rho}_{r_p}^{\pi^*}(s)] + [\hat{\rho}_{r_p}^{\tilde{\pi}^*}(s) - \hat{\rho}_{r_p}^{\hat{\pi}^*}(s)] + [\hat{\rho}_{r_p}^{\hat{\pi}^*}(s) - \hat{\rho}_{r_p}^{\hat{\pi}}(s)]$$

$$+ \hat{\rho}_{r_p}^{\hat{\pi}}(s) - \rho_r^{\hat{\pi}}(s)$$

$$= \underbrace{[\rho_r^{\pi^*}(s) - \rho_{r_p}^{\pi^*}(s)]}_{\text{Perturbation Error}} + \underbrace{[\rho_{r_p}^{\pi^*}(s) - \hat{\rho}_{r_p}^{\pi^*}(s)]}_{\text{Concentration Error}} + \underbrace{[\hat{\rho}_{r_p}^{\tilde{\pi}^*}(s) - \hat{\rho}_{r_p}^{\hat{\pi}^*}(s)]}_{\text{Sensitivity Error}} + \underbrace{[\hat{\rho}_{r_p}^{\hat{\pi}^*}(s) - \hat{\rho}_{r_p}^{\hat{\pi}}(s)]}_{\text{Primal-Dual Error}}$$

$$+ \underbrace{[\hat{\rho}_{r_p}^{\hat{\pi}}(s) - \rho_{r_p}^{\hat{\pi}}(s)]}_{\text{Concentration Error}} + \underbrace{[\rho_{r_p}^{\hat{\pi}}(s) - \rho_r^{\hat{\pi}}(s)]}_{\text{Perturbation Error}}$$

For a perturbation magnitude equal to $\omega$, we can bound both perturbation errors by $\omega$. Using Theorem 1 to bound the primal-dual error by $\varepsilon_{\text{opt}}$,

$$\leq 2\omega + \varepsilon_{\text{opt}} + \underbrace{[\rho_{r_p}^{\pi^*}(s) - \hat{\rho}_{r_p}^{\pi^*}(s)]}_{\text{Concentration Error}} + \underbrace{[\hat{\rho}_{r_p}^{\tilde{\pi}^*}(s) - \hat{\rho}_{r_p}^{\hat{\pi}^*}(s)]}_{\text{Sensitivity Error}} + \underbrace{[\hat{\rho}_{r_p}^{\hat{\pi}}(s) - \rho_{r_p}^{\hat{\pi}}(s)]}_{\text{Concentration Error}}$$

Since $b' = b + \Delta$ and setting $b'' = b - \Delta$, we use Lemma 15 to bound the sensitivity error term,

$$\rho_r^{\pi^*}(s) - \rho_r^{\hat{\pi}}(s) \leq 2\omega + \varepsilon_{\text{opt}} + 2\Delta\lambda^* + \underbrace{[\rho_{r_p}^{\pi^*}(s) - \hat{\rho}_{r_p}^{\pi^*}(s)]}_{\text{Concentration Error}} + \underbrace{[\hat{\rho}_{r_p}^{\hat{\pi}}(s) - \rho_{r_p}^{\hat{\pi}}(s)]}_{\text{Concentration Error}}$$

With these values of $b'$ and $b''$, we require the following statements to hold,

$$\left| \rho_c^{\hat{\pi}}(s) - \hat{\rho}_c^{\hat{\pi}}(s) \right| \leq \Delta - \varepsilon_{\text{opt}} \quad ; \quad \left| \rho_c^{\pi^*}(s) - \hat{\rho}_c^{\pi^*}(s) \right| \leq \Delta.$$

$\square$

# D CONCENTRATION PROOFS

**Lemma 11** (From AMDP to DMDP). *Set $\gamma = 1 - \frac{\varepsilon_{opt}}{4(B+H)}$. If the concentration error for the discounted MDP satisfies $\|V_\gamma^\pi - \hat{V}_\gamma^\pi\|_\infty \le B + H$, then it follows that $\|\rho^\pi - \hat{\rho}^\pi\|_\infty \le \varepsilon_{opt}$.*

*Proof.* We begin by decomposing the error term:

$$\frac{1}{1-\gamma}\|\rho^\pi - \hat{\rho}^\pi\|_\infty \le \|V_\gamma^\pi - \hat{V}_\gamma^\pi\|_\infty + \left\|V_\gamma^\pi - \tfrac{1}{1-\gamma}\rho^\pi\right\|_\infty + \left\|\hat{V}_\gamma^\pi - \tfrac{1}{1-\gamma}\hat{\rho}^\pi\right\|_\infty. \tag{26}$$

The first term in (26) is bounded by assumption:

$$\|V_\gamma^\pi - \hat{V}_\gamma^\pi\|_\infty \le B + H.$$

The second term can be bounded using Lemma 12, which yields

$$\left\|V_\gamma^\pi - \tfrac{1}{1-\gamma}\rho^\pi\right\|_\infty \le H.$$

Similarly, we can bound the empirical error between average and discounted setting by

$$\left\|\hat{V}_\gamma^\pi - \tfrac{1}{1-\gamma}\hat{\rho}^\pi\right\|_\infty \le 2H,$$

with only a sample complexity independent of $\varepsilon$. Combining these bounds, we obtain

$$\frac{1}{1-\gamma}\|\rho^\pi - \hat{\rho}^\pi\|_\infty \le (B + H) + H + 2H = B + 4H.$$

Now, setting

$$\gamma = 1 - \frac{\varepsilon_{\text{opt}}}{4(B+H)},$$

implies that

$$\|\rho^\pi - \hat{\rho}^\pi\|_\infty \le \varepsilon_{\text{opt}},$$

which concludes the proof. $\qquad\square$

**Lemma 12.** *We have*

$$\|V_\gamma^\pi - \frac{1}{1-\gamma}\rho^\pi\|_\infty \le H.$$

*Proof.* We begin by observing that $\pi$ satisfies

$$\rho^\pi + h^\pi = r_\pi + P_\pi h^\pi.$$

Therefore, it holds that

$$\begin{aligned}
V_\gamma^\pi &= (I - \gamma P_\pi)^{-1} r_\pi \\
&= (I - \gamma P_\pi)^{-1}(\rho^\pi + h^\pi - P_\pi h^\pi) \\
&= (I - \gamma P_\pi)^{-1}\rho^\pi + (I - \gamma P_\pi)^{-1}(I - P_\pi)h^\pi.
\end{aligned}$$

Since $P_\pi \rho^\pi = \rho^\pi$, we can calculate that

$$(I - \gamma P_\pi)^{-1}\rho^\pi = \sum_{t\ge 0}\gamma^t P_\pi^t \rho^\pi = \sum_{t\ge 0}\gamma^t \rho^\pi = \frac{1}{1-\gamma}\rho^\pi.$$

It also holds that

$$\begin{aligned}
(I - \gamma P_\pi)^{-1}(I - P_\pi) &= \sum_{t\ge 0}\gamma^t P_\pi^t(I - P_\pi) \\
&= \sum_{t\ge 0}\gamma^t P_\pi^t - \sum_{t\ge 0}\gamma^t P_\pi^{t+1} \\
&= P_\pi + \sum_{t\ge 0}(\gamma^{t+1} - \gamma^t)P_\pi^{t+1} \tag{27}
\end{aligned}$$

and $\sum_{t\ge 0}\gamma^{t+1} - \gamma^t = (\gamma - 1)\sum_{t\ge 0}\gamma^t = -1$. Therefore (27) is the difference of two stochastic matrices, and so it follows that

$$\|(I - \gamma P_\pi)^{-1}(I - P_\pi)h^\pi\|_\infty \le H.$$

$\qquad\square$

**Lemma 13** (Theorem 6 of Vaswani et al. (2022)). *For $\delta \in (0,1)$, $\omega \leq 1$ and $C(\delta) = 72 \log\left(\frac{16(1+U+\omega)\,SA\log(e/1-\gamma)}{(1-\gamma)^2\,\iota\,\delta}\right)$ where $\iota = \frac{\omega\,\delta\,(1-\gamma)\,\varepsilon_l}{30\,U|S||A|^2}$, if $N \geq \frac{4\,C(\delta)}{1-\gamma}$, then for $\hat{\pi}$ output by Algorithm 1, with probability at least $1 - \delta/5$,*

$$\left|V_{r_p}^{\hat{\pi}}(s) - \hat{V}_{r_p}^{\hat{\pi}}(s)\right| \leq 2\sqrt{\frac{C(\delta)}{N \cdot (1-\gamma)^3}} \quad ; \quad \left|V_c^{\hat{\pi}}(s) - \hat{V}_c^{\hat{\pi}}(s)\right| \leq \sqrt{\frac{C(\delta)}{N \cdot (1-\gamma)^3}}.$$

**Lemma 14** (Lemma 7 of Vaswani et al. (2022)). *For $\delta \in (0,1)$, $\omega \leq 1$ and $C'(\delta) = 72 \log\left(\frac{4|S|\log(e/1-\gamma)}{\delta}\right)$, if $N \geq \frac{4\,C'(\delta)}{1-\gamma}$ and $B(\delta, N) := \sqrt{\frac{C'(\delta)}{(1-\gamma)^3 N}}$, then with probability at least $1 - 3\delta$,*

$$\left|V_{r_p}^{\pi^*}(s) - \hat{V}_{r_p}^{\pi^*}(s)\right| \leq 2B(\delta, N); \left|V_c^{\pi^*}(s) - \hat{V}_c^{\pi^*}(s)\right| \leq B(\delta, N); \left|V_c^{\pi_c^*}(s) - \hat{V}_c^{\pi_c^*}(s)\right| \leq B(\delta, N)$$

# E SUPPORTING LEMMAS FOR THE UPPER BOUND

**Lemma 15** (Bounding the sensitivity error). *If $b' = b + \Delta$ such that,*

$$\hat{\pi}^* \in \arg\max_{\pi} \rho_r^{\pi}(s) \text{ s.t. } \rho_c^{\pi}(s) \geq b + \Delta$$

$$\pi^* \in \arg\max_{\pi} \rho_r^{\pi}(s) \text{ s.t. } \rho_c^{\pi}(s) \geq b,$$

*then the sensitivity error term can be bounded by:*

$$\left|\rho_r^{\hat{\pi}^*}(s) - \rho_r^{\pi^*}(s)\right| \leq \Delta\lambda^*.$$

*Proof.* Writing the reference CAMDP in Eq. (4) in its Lagrangian form,

$$\rho_r^{\hat{\pi}^*}(s) = \max_{\pi} \min_{\lambda \geq 0} \rho_r^{\pi}(s) + \lambda[\rho_c^{\pi}(s) - (b + \Delta)]$$

$$= \min_{\lambda \geq 0} \max_{\pi} \rho_r^{\pi}(s) + \lambda[\rho_c^{\pi}(s) - (b + \Delta)] \qquad \text{(By strong duality Lemma 6)}$$

Since $\lambda^*$ is the optimal dual variable for the empirical CMDP in Eq. (4),

$$= \max_{\pi} \rho_r^{\pi}(s) + \lambda^*[\rho_c^{\pi}(s) - (b + \Delta)]$$

$$\geq \rho_r^{\pi^*}(s) + \lambda^*[\rho_c^{\pi^*}(s) - (b + \Delta)] \qquad \text{(The relation holds for } \pi = \pi^*.)$$

Since $\rho_c^{\pi^*}(s) \geq b$,

$$\rho_r^{\hat{\pi}^*}(s) \geq \rho_r^{\pi^*}(s) - \lambda^*\Delta$$

$$\implies \rho_r^{\pi^*}(s) - \rho_r^{\hat{\pi}^*}(s) \leq \Delta\lambda^*$$

Since the CAMDP with $b' = b$ is a less constrained problem than the one in Eq. (4) (with $b' = b + \Delta$), $\rho_r^{\pi^*}(s) \geq \rho_r^{\hat{\pi}^*}(s)$, and hence,

$$\left|\rho_r^{\pi^*}(s) - \rho_r^{\hat{\pi}^*}(s)\right| \leq 2\Delta\lambda^*.$$

$\square$

**Lemma 16** (Bounding the optimal bias function and the transient time). *If the AMDPs $(\mathcal{P}, r)$ and $(\mathcal{P}, c)$ admit bias functions bound with parameter $H$ and satisfy the bounded transient time assumption with parameter $B$, then the combined AMDP $(\mathcal{P}, r + \lambda c)$, where $\lambda$ is as defined in Eq. (6), also satisfies the bounded bias functions assumption with parameter $H$ and the bounded transient time assumption with parameter $B$, after normalizing the reward values.*

*Proof.* Based on the bounded transient time assumption, for all $\pi \in \Pi$ and $s \in \mathcal{S}$, we have

$$\mathbb{E}_s^{\pi}[T_{\mathcal{R}^{\pi}}] \leq B, \quad \text{where} \quad T_{\mathcal{R}^{\pi}} := \inf\{t \geq 0 : S_t \in \mathcal{R}^{\pi}\}.$$

Since the transient time parameter $B$ is determined solely by the transition dynamics of the AMDP and is independent of the reward function, it follows that the combined AMDP $(\mathcal{P}, r + \lambda c)$ also satisfies the bounded transient time assumption.

We now turn to bounding the span of the optimal bias function under the combined reward $r + \lambda c$. Let $\pi^*$ denote the optimal policy for this reward. By linearity of the bias operator with respect to reward and the definition of span, we have

$$
\begin{aligned}
\left\| h^*_{r+\lambda c} \right\|_{\text{span}} &= \frac{1}{1+\lambda} \left\| \underset{T \to \infty}{\text{C-lim}} \, \mathbb{E}_s^{\pi^*} \left[ \sum_{t=0}^{T-1} \left( r_t + \lambda c_t - \rho_r^{\pi^*} - \lambda \rho_c^{\pi^*} \right) \right] \right\|_{\text{span}} \\
&\leq \frac{1}{1+\lambda} \left( \left\| \underset{T \to \infty}{\text{C-lim}} \, \mathbb{E}_s^{\pi^*} \left[ \sum_{t=0}^{T-1} \left( r_t - \rho_r^{\pi^*} \right) \right] \right\|_{\text{span}} + \lambda \left\| \underset{T \to \infty}{\text{C-lim}} \, \mathbb{E}_s^{\pi^*} \left[ \sum_{t=0}^{T-1} \left( c_t - \rho_c^{\pi^*} \right) \right] \right\|_{\text{span}} \right) \\
&= \frac{H + \lambda H}{1 + \lambda} \\
&\leq H,
\end{aligned}
$$

$\square$

---

**Lemma 17** (Sample Complexity to Estimate Bias Span). *Let $H$ denote the bias-span parameter, $H := \max_\pi \| h^\pi \|_{\text{span}} = \max_\pi \left( \max_s h^\pi(s) - \min_s h^\pi(s) \right)$. Then, under access to a generative model, the quantity $H$ can be estimated to constant-factor accuracy using $\widetilde{O}(SAD)$ samples.*

---

*Proof.* Fix a reference state $s_0$ in a recurrent class of $\pi$ and normalize the bias so that $h^\pi(s_0) = 0$. For any state $s$, consider the trajectory obtained by starting from $s$, following $\pi$, and stopping when the chain hits $s_0$ for the first time. Let $T_s$ be this hitting time and define the random variable

$$
Z_s := \sum_{t=0}^{T_s - 1} \left( r(s_t, \pi(s_t)) - \rho_r^\pi(s_t) \right),
$$

where $\rho_r^\pi$ is the (state-dependent) average reward vector under $\pi$. By standard average-reward theory, we have $\mathbb{E}[Z_s] = h^\pi(s) - h^\pi(s_0) = h^\pi(s)$.

Each trajectory length $T_s$ is at most $D$ in expectation, and every increment $r(s_t, \pi(s_t)) - \rho_r^\pi(s_t)$ is bounded in $[-1, 1]$. Thus $Z_s$ has magnitude and variance on the order of $D$ and $D^2$, respectively. To estimate $\mathbb{E}[Z_s]$ up to additive error $\alpha H$ for some fixed small constant $\alpha \in (0,1)$, Bernstein-type concentration inequalities imply that a constant number $n_s = \tilde{O}(1)$ of independent trajectories starting from $s$ suffice: the target accuracy $\alpha H$ is of the same order as the typical size of $Z_s$, so only $O(1)$ samples are needed to obtain a constant-factor estimate. Each such trajectory requires $\tilde{O}(D)$ environment interactions in expectation, so the sample cost per state is $\tilde{O}(D)$.

Repeating this construction for all $SA$ state–action pairs) and applying a union bound, we obtain an estimator $\hat{h}^\pi$ such that

$$
\max_s \left| \hat{h}^\pi(s) - h^\pi(s) \right| \leq \alpha H
$$

with high probability. Consequently, the empirical bias span $\hat{H} := \max_\pi \| \hat{h}^\pi \|_{\text{span}}$ satisfies

$$
\left| \hat{H} - H \right| \leq 2\alpha H, \qquad \text{and hence} \qquad \hat{H} \leq (1 + 2\alpha) H,
$$

so $\hat{H}_\pi$ is a constant-factor upper bound on $H_\pi$. The total number of environment interactions used is $\tilde{O}(SAD)$.

$\square$

---

**Lemma 18** (Sample Complexity to Estimate Transient Time Bound). *Let $B$ be the transient time bound defined as $\forall \pi, s, \quad \mathbb{E}_s^\pi[T_{\mathcal{R}^\pi}] \leq B$, where $T_{\mathcal{R}^\pi}$ is the first hitting time to a recurrent state under policy $\pi$. Then, under access to a generative model or an environment where episodes can be reset to any state-action pair, the transient time bound $B$ can be estimated up to a constant-factor accuracy using $\widetilde{O}(SAB)$ samples.*

---

*Proof.* To estimate the expected hitting time $\mathbb{E}_s^\pi[T_{\mathcal{R}^\pi}]$ from each state $s$ under a fixed policy $\pi$, we sample full trajectories until they reach the recurrent class $\mathcal{R}^\pi$. Each trajectory is a random variable $T \in \mathbb{N}$ with expectation at most $B$ and variance $\text{Var}(T) = O(B^2)$.

To estimate $\mathbb{E}[T]$ up to additive error $\varepsilon = \Theta(B)$, standard concentration inequalities (e.g., Bernstein's inequality) imply that

$$n = O\left(\frac{B^2 \log(1/\delta)}{\varepsilon^2}\right) = \widetilde{O}(1)$$

trajectories suffice.

Each trajectory requires $\Theta(B)$ environment interactions in expectation, so the sample cost per state-action pair is $\widetilde{O}(B)$. Summing over all $SA$ state-action pairs yields a total sample complexity of

$$\widetilde{O}(SAB).$$

$\square$

## F PROOFS FOR LOWER-BOUND FOR GENERAL CAMDPS

**Theorem 5** (Lower-bound for general CAMDP). *For any sufficiently small $\varepsilon, \delta$, any sufficiently large $S, A$, for any algorithm promising to return an $\frac{\varepsilon}{24}$-optimal policy with probability at least $\frac{3}{4}$ on any communicating CAMDP problem, there is a CAMDP such that the expected total samples on all state-action pairs, when running this algorithm, is at least $\widetilde{\Omega}\left(\frac{SA(H+B)}{\varepsilon^2 \zeta^2}\right)$*

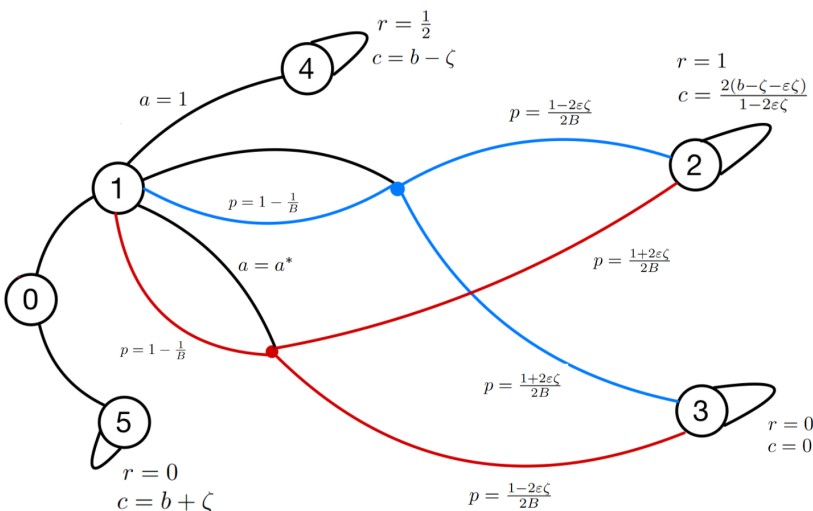

Figure 4: A Component MDP Used in the Hard Instance for CAMDP.

*Proof.* We begin by introducing a family of MDP instances $M_{a^*}$ indexed by $a^* \in \{1, \ldots, A\}$, depicted in Figure Fig. 4. In all these instances, states $2, 3, 4, 5$ are absorbing, while states $0$ and $1$ are transient. Among them, state $1$ is the only one with multiple actions, supporting $A$ distinct actions. Taking action $a = 1$ from state $1$ deterministically leads to state $4$. For action $a = 2$, the transition probabilities are defined as $P(1 \mid 1, 2) = 1 - \frac{1}{B}$, $P(2 \mid 1, 2) = p_2$, and $P(3 \mid 1, 2) = 1 - P(1 \mid 1, 2) - P(2 \mid 1, 2)$. The specific values of $P(2 \mid 1, a)$, $P(3 \mid 1, a)$, and the reward and constraint values $r$ and $c$ are shown in Figure 4, and are the only quantities that vary across the different instances $M_{a^*}$. Note that all actions not in state $1$ can only lead to one state.

In instance $M_1$, the optimal policy selects action $a = 1$, achieving an average reward of $1/2$. Choosing any other action results in a suboptimal average reward of $\frac{1-2\varepsilon\zeta}{2}$. For instances $M_{a^*}$ with $a^* \in \{2, \ldots, A\}$, the optimal action is $a = a^*$, yielding an average reward of $\frac{1+2\varepsilon\zeta}{2}$, while action $a = 1$ returns $\frac{1}{2}$, and all remaining actions incur a reward of $\frac{1-2\varepsilon\zeta}{2}$. In all such cases, the span of the bias function under the optimal policy satisfies $\|h^*\|_{\text{span}} = 0$. An analogous construction holds for the constraint rewards $c$. Furthermore, any action $a \neq 1$ leads the agent to remain in state $1$ for an expected $B$ steps before transitioning to either state $2$ or $3$, thus ensuring that the bounded transient time condition is met with parameter $B$.

We then define a set of $(A-1)S/6$ *master MDPs* denoted $M_{s^*,a^*}$, indexed by $s^* \in \{1,\ldots,S/6\}$ and $a^* \in \{2,\ldots,A\}$. Each master MDP consists of $S/6$ independent copies of the sub-MDPs described above, which are all connected to an initial state. The $s^*$-th sub-MDP is set to be $M_{a^*}$, while all remaining sub-MDPs are instantiated as $M_1$. To ensure non-overlapping state spaces, the states of the $s$-th sub-MDP are relabeled as $6s, 6s+1, \ldots, 6s+5$, corresponding to states $0, 1, \ldots, 5$ in Figure 4. We also define $M_0$ composed of $S/6$ independent $M_1$. As a result, each master MDP has exactly $S$ states and $A$ actions, satisfies the bounded transient time condition with parameter $B$, and possesses an optimal policy with bias span zero.

We further fix the constraint threshold to $b = \frac{1}{2}$ in the construction of our hard CAMDP instances. Based on the structure depicted in Fig. 4, we directly compute the expected reward and constraint values as follows: in states of the form $6s+1$, choosing action $a_1$ yields reward $r = \frac{1}{2}$ and constraint value $c = b - \zeta$, while selecting any action $a \in \mathcal{A} \setminus \{a_1\}$ results in reward $r = \frac{1}{2} - \varepsilon\zeta$ and constraint $c = b - \zeta - \varepsilon\zeta$.

At the special state $6s^* + 1$, the designated optimal action $a^*$ yields reward $r = \frac{1}{2} + \varepsilon\zeta$, and the corresponding constraint value is given by $c = \frac{(b-\zeta-\varepsilon\zeta)(1+2\varepsilon\zeta)}{1-2\varepsilon\zeta} = b - \zeta + \varepsilon\zeta - 4\varepsilon\zeta^2 + o(\varepsilon)$.

Let $s_0$ denote the initial state that connects to all branches $6s$, and define the following occupancy measures:

- $\mu_0 = \sum_{s=0}^{S/6-1} p(s_0, 6s) \cdot p(s, 6s+5)$,

- $\mu_1 = \sum_{s=0}^{S/6-1} p(s_0, 6s) \cdot p(6s, 6s+1) \cdot p(6s+1, a_1)$,

- $\mu_2 = \sum_{a \in A} \sum_{s=0}^{S/6-1} p(s_0, 6s) \cdot p(6s, 6s+1) \cdot p(6s+1, a)$ for $a \in \mathcal{A} \setminus \{a_1\}$.

We now formulate the linear program (LP) for solving the average-reward objective in $M_0$:

$$\max \quad \frac{1}{2}\mu_1 + \left(\frac{1}{2} - \varepsilon\zeta\right)\mu_2 \tag{28}$$
$$\text{s.t.} \quad \mu_0 + \mu_1 + \mu_2 = 1,$$
$$(b+\zeta)\mu_0 + (b-\zeta)\mu_1 + (b-\zeta-\varepsilon\zeta)\mu_2 \geq b,$$
$$\mu_0, \mu_1, \mu_2 \geq 0.$$

The unique optimal solution to Eq. (28) is $\mu_0 = \frac{1}{2}$, $\mu_1 = \frac{1}{2}$, and $\mu_2 = 0$, yielding an average reward $\rho^*(s_0) = \frac{1}{4}$.

Next, we aim to show that for any $\frac{\varepsilon}{24}$-optimal policy, the normalized occupancy $\mu_1' := \frac{\mu_1}{1-\mu_0}$ must satisfy $\mu_1' \geq \frac{2}{3}$. Suppose, for contradiction, that $\mu_1' < \frac{2}{3}$. The modified LP becomes:

$$\max \quad \frac{1}{2}\mu_1 + \left(\frac{1}{2} - \varepsilon\zeta\right)\mu_2 \tag{29}$$
$$\text{s.t.} \quad \mu_0 + \mu_1 + \mu_2 = 1, \quad \mu_1' < \frac{2}{3},$$
$$(b+\zeta)\mu_0 + (b-\zeta)\mu_1 + (b-\zeta-\varepsilon\zeta)\mu_2 \geq b,$$
$$\mu_0, \mu_1, \mu_2 \geq 0.$$

A direct calculation shows that the optimal reward for Eq. (29) is $\rho(s_0) = \frac{1}{4} - \frac{\varepsilon}{24} - \frac{\varepsilon\zeta}{6}$, which violates the $\frac{\varepsilon}{24}$-optimality condition. Therefore, the assumption $\mu_1' < \frac{2}{3}$ must be false, and it follows that any $\frac{\varepsilon}{24}$-optimal policy must satisfy $\mu_1' \geq \frac{2}{3}$.

For CAMDP $M_{s^*,a^*}$, we define the two new occupancy measures:

- $\mu_2^c = \mu_2 - p(s_0, 6s^*) \cdot p(6s^*, 6s^*+1) \cdot p(6s^*+1, a*)$.

- $\mu_3 = p(s_0, 6s^*) \cdot p(6s^*, 6s^*+1) \cdot p(6s^*+1, a*)$

We now formulate the LP for solving the average-reward objective in $M_{s^*,a^*}$:

$$\max \quad \frac{1}{2}\mu_1 + \left(\frac{1}{2} - \varepsilon\zeta\right)\mu_2^c + \left(\frac{1}{2} + \varepsilon\zeta\right)\mu_3 \tag{30}$$

$$\text{s.t.} \quad \mu_0 + \mu_1 + \mu_2^c + \mu_3 = 1,$$

$$(b+\zeta)\mu_0 + (b-\zeta)\mu_1 + (b-\zeta-\varepsilon\zeta)\mu_2^c + [b - \zeta + \varepsilon\zeta - 4\varepsilon\zeta^2 + o(\varepsilon)]\mu_3 \geq b,$$

$$\mu_0, \mu_1, \mu_2^c, \mu_3 \geq 0.$$

The unique optimal solution to Eq. (30) is $\mu_0 = \frac{1+\varepsilon-\varepsilon\zeta}{2-\varepsilon+\varepsilon\zeta} + o(\varepsilon)$, $\mu_1 = \mu_2^c = 0$, $\mu_3 = \frac{1}{2-\varepsilon+\varepsilon\zeta} + o(\varepsilon)$ yielding an average reward $\rho^*(s_0) = \frac{1}{4} + \frac{\varepsilon}{8} + \frac{3\varepsilon\zeta}{8} + o(\varepsilon)$.

Next, we aim to show that for any $\frac{\varepsilon}{24}$-optimal policy, the normalized occupancy $\mu_1'$ must satisfy $\mu_1' \leq \frac{2}{3}$. Suppose, for contradiction, that $\mu_1' > \frac{2}{3}$. The modified LP becomes:

$$\max \quad \frac{1}{2}\mu_1 + \left(\frac{1}{2} - \varepsilon\zeta\right)\mu_2^c + \left(\frac{1}{2} + \varepsilon\zeta\right)\mu_3 \tag{31}$$

$$\text{s.t.} \quad \mu_0 + \mu_1 + \mu_2^c + \mu_3 = 1, \mu_1' > \frac{2}{3}$$

$$(b+\zeta)\mu_0 + (b-\zeta)\mu_1 + (b-\zeta-\varepsilon\zeta)\mu_2^c + [b - \zeta + \varepsilon\zeta - 4\varepsilon\zeta^2 + o(\varepsilon)]\mu_3 \geq b,$$

$$\mu_0, \mu_1, \mu_2^c, \mu_3 \geq 0.$$

A direct calculation shows that the optimal reward for Eq. (31) is $\rho(s_0) = \frac{1}{4} + \frac{\varepsilon}{24} + o(\varepsilon)$, which violates the $\frac{\varepsilon}{24}$-optimality condition. Therefore, the assumption $\mu_1' > \frac{2}{3}$ must be false, and it follows that any $\frac{\varepsilon}{24}$-optimal policy must satisfy $\mu_1' \leq \frac{2}{3}$.

In short, for any $\frac{\varepsilon}{24}$-optimal policy, $\mu_1'$ must satisfy $\mu_1' \leq \frac{2}{3}$ for $M_{s^*,a^*}$ and $\mu_1' \geq \frac{2}{3}$ for $M_0$.

So we can use the Fano's method to lower bound the failure probability. We have:

$$P_{M_{s^*,a^*}}\left(\cdot \mid 6s^* + 1, a^*\right) = \mathrm{Cat}\left(1 - \frac{1}{B}, \frac{1 - 2\varepsilon\zeta}{2B}, \frac{1 + 2\varepsilon\zeta}{2B}\right) =: Q_1,$$

$$P_{M_0}\left(\cdot \mid 6s^* + 1, a^*\right) = \mathrm{Cat}\left(1 - \frac{1}{B}, \frac{1 + 2\varepsilon\zeta}{2B}, \frac{1 - 2\varepsilon\zeta}{2B}\right) =: Q_2,$$

where $\mathrm{Cat}(p_1, p_2, p_3)$ denotes the categorical distribution with event probabilities $p_i$'s.

Now we use Fano's method to lower bound this failure probability. This is inspired by the proof of lower-bound for AMDP in Zurek & Chen (2024). Choose an index $J$ uniformly at random from the set $\mathcal{J} := \{1, \ldots, S/6\} \times \{2, \ldots, A\}$ and suppose that we draw $n$ iid samples $X = (X_1, \ldots, X_n)$ from the master MDP $M_J$; note that under the generative model, each random variable $X_i$ represents an $(S \times A)$-by-$S$ transition matrix with exactly one nonzero entry in each row. Letting $\mathrm{I}(J; X)$ denote the mutual information between $J$ and $X$, Fano's inequality yields that the failure probability is lower bounded by

$$1 - \frac{\mathrm{I}(J; X) + \log 2}{\log((A-1)S/6)}.$$

We can calculate using the fact that the $P_i$'s are i.i.d., the chain rule of mutual information, and the form of the construction that

$$\mathrm{I}(J; X) = n\mathrm{I}(J; X_1)$$

$$\leq n \max_{(s^*,a^*)\in\mathcal{J}} \mathrm{D_{KL}}\left(P_{M_{s^*,a^*}} \mid P_{M_0}\right)$$

$$= n\mathrm{D_{KL}}(Q_1 \mid Q_2).$$

By direct calculation, we have

$$
\begin{aligned}
\mathrm{D}_{\mathrm{KL}}(Q_1|Q_2) &= \frac{1-2\varepsilon\zeta}{2B}\log\frac{1-2\varepsilon\zeta}{1+2\varepsilon\zeta} + \frac{1+2\varepsilon\zeta}{2B}\log\frac{1+2\varepsilon\zeta}{1-2\varepsilon\zeta} \\
&\leq \frac{1-2\varepsilon\zeta}{2B}\cdot\frac{-4\varepsilon\zeta}{1+2\varepsilon\zeta} + \frac{1+2\varepsilon\zeta}{2B}\cdot\frac{4\varepsilon\zeta}{1-2\varepsilon\zeta} \qquad \log(1+x)\leq x, \forall x > -1 \\
&= \frac{16\varepsilon^2\zeta^2}{B(1+2\varepsilon\zeta)(1-2\varepsilon\zeta)} \\
&\leq \frac{32\varepsilon^2\zeta^2}{B} \qquad\qquad\qquad\qquad\qquad\qquad \varepsilon\zeta\leq\frac{1}{4}.
\end{aligned}
$$

Therefore the failure probability is at least

$$
1 - \frac{\mathrm{I}(J;P^n)+\log 2}{\log((A-1)S/6)} \geq 1 - \frac{n\frac{32\varepsilon^2\zeta^2}{B}+\log 2}{\log((A-1)S/6)}
$$

$$
\geq \frac{1}{2} - \frac{n\frac{32\varepsilon^2\zeta^2}{B}}{\log((A-1)S/6)},
$$

where in the second inequality we assumed $A$ and $S$ are at least a sufficiently large constant. For the above RHS to be smaller than $1/4$, we therefore require $n \geq \tilde{\Omega}(\frac{B\log(SA)}{\varepsilon^2\zeta^2})$. Finally, by combining this result with Theorem 4, we obtain the general lower bound for general CAMDPs: $\tilde{\Omega}\left(\frac{SA(B+H)}{\varepsilon^2\zeta^2}\right)$. $\quad\square$

## G  PROOFS FOR LOWER-BOUND FOR WEAKLY COMMUNICATING CAMDPS

**Theorem 4** (Lower-bound for communicating CAMDP). *For any sufficiently small $\varepsilon$, $\delta$, any sufficiently large $S$, $A$, and any $D \geq \max\{c_1 S, c_2\}$ (where $c_1, c_2 \geq 0$ is some universal constant), for any algorithm promising to return an $\frac{\varepsilon}{24}$-optimal policy with probability at least $\frac{3}{4}$ on any communicating CAMDP problem, there is a CAMDP such that the expected total samples on all state-action pairs, when running this algorithm, is at least $\tilde{\Omega}\left(\frac{SAH}{\varepsilon^2\zeta^2}\right)$*

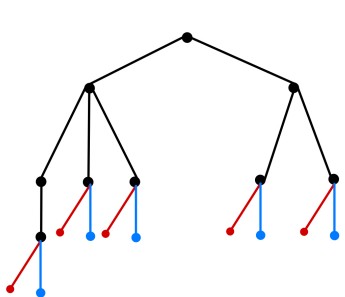

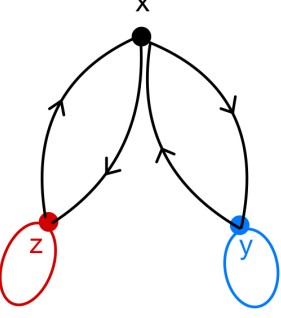

Figure 5: A Hard Communicating CAMDP when $A = 4$, $S = 19$.

Figure 6: A Component Communicating CAMDP.

*Proof.* To construct a family of hard MDP instances with parameters $S$, $A$ and diameter at most $D$, we begin by introducing key components and associated notation. Define $A' := A - 1$, $D' := D/8$, and $K := \lceil S/4 \rceil$. We assume that $A \geq 3$, $\varepsilon \leq 1/16$, and $D \geq \max\{16\lceil\log_A S\rceil, 16\}$, which are standard parameter ranges in this construction.

We first define a primitive component MDP consisting of three states $x, y, z$, each equipped with $A'$ actions and parameterized by $D'$. The action space is partitioned into three subsets based on their transition and reward behavior. This component MDP serves as a key building block in the lower bound construction and is illustrated in Figure 6.

Next, we assemble $K$ identical copies of the component MDP into a larger structure $M_0$, which serves as the base instance for constructing the lower bound family. We begin by constructing an $A'$-ary rooted tree with exactly $S - 3K$ non-leaf nodes and $K$ leaves. It is known that such a tree exists with depth at most $\lceil \log_{A'} S \rceil + 1$. Each leaf of this tree is replaced by a component MDP: the node corresponding to the leaf becomes state $x$, while its two children are mapped to states $y$ and $z$. The final MDP $M_0$ is thus formed by embedding the component MDPs into the leaf structure of the tree, as illustrated in Figure 5.

Transitions in the tree are defined deterministically: every internal node (including $x$-nodes) has actions that lead to each of its children and its parent (if applicable); all remaining actions correspond to self-loops with zero reward. For each $y$-state in the embedded components, one designated action is also a deterministic self-loop with zero reward. By construction, $K \geq S/4$, and the overall diameter of $M_0$ is bounded as: $2 \left( \frac{D'}{1+8\varepsilon} + \log_{A'} S + 1 \right) \leq D$, given the definition $D' := D/8$ and the assumed bound $\log_A S \leq D/8$.

We then define a collection of hard instances $\{M_{k,l}\}_{1 \leq k \leq K, \, 2 \leq l \leq A'}$ based on perturbations of $M_0$. To distinguish among these instances, note that a policy must favor action $a_1$ at the $x_k$ states in $M_0$, while selecting $a_l$ in the corresponding $M_{k,l}$. Specifically, to be $\varepsilon/24$-optimal in $M_{k,l}$, the policy must assign occupancy measure at most $2/3$ to action $a_1$ at state $x_k$, while in $M_0$, the same state must have occupancy measure at least $2/3$ on $a_1$. This divergence in action distributions under different instances forms the basis of our lower bound. The design of our hard instance is motivated by the construction used for average-reward MDPs in Wang et al. (2022).

We further fix the constraint threshold to $b = \frac{1}{2}$ in the construction of our hard CAMDP instances (Figure 5). Building on the analysis in Appendix F, we leverage a carefully designed reward and constraint structure to induce a separation in policy behavior across different MDP instances.

Under our construction, we can show that any policy that is $\frac{\varepsilon}{24}$-optimal must satisfy distinct occupancy conditions across instances: in the base instance $M_0$, the normalized occupancy measure $\mu'_1$–representing the fraction of trajectories where action $a_1$ is selected–must satisfy $\mu'_1 \geq \frac{2}{3}$; in contrast, for any perturbed instance $M_{k,l}$, the same quantity must satisfy $\mu'_1 \leq \frac{2}{3}$. This divergence in occupancy thresholds arises due to the amplification effect in the constraint values, and ensures that policies achieving small regret in one instance must necessarily incur significant suboptimality in others.

This behavioral separation enables us to apply Fano's method to formally lower bound the probability of misidentifying the underlying instance. Following the same framework as in Appendix F, we derive a lower bound on the sample complexity of learning an $\varepsilon$-optimal policy under strict feasibility: $\tilde{\Omega} \left( \frac{SAD}{\varepsilon^2 \zeta^2} \right)$. Furthermore, by noting that the bias span $H$ is always bounded above by the diameter $D$, this implies a corresponding lower bound of $\tilde{\Omega} \left( \frac{SAH}{\varepsilon^2 \zeta^2} \right)$, which holds for the class of weakly communicating constrained average-reward MDPs. $\qquad\square$

## STATEMENT OF LLM USAGE

This manuscript used large language models solely to assist with language editing and improving the clarity of writing. All technical content, analysis, and conclusions were conceived, implemented, and verified entirely by the authors.

