# OpenReview forum: "Near-Optimal Sample Complexity Bounds for Constrained Average-Reward MDPs"
_ICLR.cc/2026/Conference — ICLR 2026 Poster_

### Official Review · Reviewer_6HuV · 2025-10-31

**Soundness:** 4
**Presentation:** 4
**Contribution:** 3
**Rating:** 6
**Confidence:** 3

**Summary:**

This paper introduce a minimax-optimal primal-dual method for constrained AMDPs under the generative setting.

**Strengths:**

The analysis is solid and close the gap in the current literature of constrained average-reward MDPs. The approach used makes sense, converting to a MDP and update the dual variable, following the standard primal-dual setup. it is impressive to see that the proposed algorithm match the lower bound, which is also presented in this paper. Overall I find this work substantial and the paper is nicely written.

**Weaknesses:**

1. As this paper proposes to use a generative model, a natural question would be, how feasible is it to extend to the markovian model setting? Oftentimes, e.g. in robotics, we cannot sample arbitrary state-action pairs even given a simulator. Thus the generative model assumption does not hold here.
2. Although the approach of reducing the AMDP to DMDP with Blackwell-optimality appears in other works, I wonder if one can solve the AMDP directly. Experience shows that large discount factor pose instability to the algorithm, limiting the applicability of the proposed framework. As such, the contribution of the work is limited to algorithm analysis.
3. I am not familiar with the field of constrained MDPs so I encourage the authors to present related work's results, even in discounted setting (as this work use some sort of discounted setup) to give a fair comparison.
4. I find the use of model-based approach failing to attract my interest, as real life problems are unlikely be handled through world model learning. In addition, for large state or action spaces, function approximation seems to be unavoidable, in which this work does not address either.
Nevertheless, I think this is solid work, for which I lean positively.

**Questions:**

See above.

---

> ### Author Response · Authors · 2025-11-20
>
> We sincerely thank the reviewer for the very positive evaluation, in particular for noting that the analysis is solid, that the paper closes a gap in the literature on constrained average-reward MDPs, and that the algorithm and lower bound match. We now address your concerns point by point.
>
> 1.
> Our focus in this work is on the statistical limits of learning constrained average-reward MDPs, and for this purpose the generative model is the standard oracle in the sample-complexity literature. It isolates the estimation problem from exploration issues and yields information-theoretic lower bounds that automatically apply to any weaker interaction model (including the online Markovian setting). Extending our upper bounds to online algorithms that must both explore and satisfy constraints is an important and challenging direction. It would require additional ideas, and we see this as a natural line of future work building on the present results.
>
> 2.
> The role of the AMDP-DMDP reduction in our work is purely analytical. Algorithm 1 itself operates directly on the average-reward CMDP: in each iteration it runs a primal–dual update on the empirical model with reward $r_p+\lambda_t c$, without introducing any discount factor in the implementation. The discount parameter $\gamma$ only appears in the proofs, where Blackwell optimality allows us to convert concentration bounds for average-reward AMDPs into bounds for suitably perturbed discounted MDPs and then leverage existing non-asymptotic results for the latter. Thus, the potential numerical stability issues associated with running large-$\gamma$ value-iteration or policy-iteration do not affect our algorithm. Our contribution here is to show that the AMDP-DMDP reduction leads to minimax-optimal sample complexity and offers a clean conceptual bridge between constrained average-reward and discounted settings.
>
> 3.
> We appreciate the suggestion to better position our results within the broader constrained MDP literature. Subject to the page limit, we will expand the related-work discussion to more explicitly summarize existing work on discounted and finite-horizon constrained MDPs and clarify how their assumptions and guarantees differ from our generative-model, average-reward setting. Where space permits, we will also add direct comparisons to several recent representative works, so that readers can more clearly see how our bounds complement and extend prior results.
>
> 4. Model-based approach and function approximation.
> We agree that many real-world applications involve large or continuous state spaces where function approximation and model-free methods are unavoidable, and our tabular, model-based setting does not directly address these regimes. In this paper we deliberately focus on the finite, generative-model setting in order to first pin down the fundamental sample-complexity limits for constrained average-reward MDPs. Such sharp results typically serve as a theoretical baseline for later work that incorporates approximation, richer function classes, or partial observability. We view extending our techniques to model-free algorithms and function-approximation settings as a natural and important direction for future research, building on the foundations established here.
>
> We are grateful for the reviewer’s thoughtful comments and are encouraged by the overall positive recommendation.

---

> > ### Comment · Reviewer_6HuV · 2025-11-26
> >
> > I thank the authors for the response. I have no further questions. Given the paper's status, I'd like to maintain my score.

---

### Official Review · Reviewer_4Q3K · 2025-10-31

**Soundness:** 3
**Presentation:** 3
**Contribution:** 3
**Rating:** 4
**Confidence:** 2

**Summary:**

This paper provides sample complexity bounds on learning constrained average-reward MDPs under the generative model setting. The authors propose a primal-dual framework that works for the relaxed feasibility and strict feasibility regimes. The algorithm achieves near-optimal sample complexity in both settings. Furthermore, the paper characterizes a matching lower bound for the strict feasibility setting that incorporates the Slater parameter. This shows that the sample complexity upper bound for the setting is (nearly) minimax optimal.

**Strengths:**

- The theoretical contributions, if correct, are clearly significant. Learning infinite-horizon average-reward constrained MDPs seems a harder problem than learning CMDPs in the finite-horizon setting or the discounted-reward setting. The lower bound that incorporates Slater's constant is novel, and it is very interesting that the lower bound matches the complexity upper bound.
- The proofs are based on novel ideas, not repeating existing analysis techniques. Their arguments are carefully prepared based on the theory of average-reward MDPs.

**Weaknesses:**

I have two concerns that, I hope, can be addressed or clarified by the authors. Let us discuss them as below.
- In each iteration, the algorithm requires solving an unconstrained average-reward MDP. Theoretically, as claimed in the paper, the part can be dealt with by a black-box MDP solver such as a linear programming-based method. However, it seems to important to incorporate errors incurred from the black-box MDP solver and to discuss its complexity.
- I think that the most subtle part is about arguing that the average-reward of approximate MDPs is bounded. However, I had hard time understanding the proof of Lemma 17. It is not clear why the bias function of an approximate MDP is bounded. Moreover, the average-reward of an approximate MDP remains constant for all states? I also have the following more specific questions about the proof.

&ensp; Why is it that (Lemma 8.6.4, Puterman 1994) implies $|\rho^\star - \hat \rho^\star|_\infty\leq O(\varepsilon)$?

&ensp; Why does $|| \hat h||_\infty\leq H$ hold?

**Questions:**

I have listed my questions in the weakness part above.

---

> ### Author Response · Authors · 2025-11-20
>
> We thank the reviewer for the thoughtful assessment and for recognizing the significance of obtaining matching upper and lower bounds for constrained average–reward MDPs.
>
> 1. Black-box AMDP solver and its complexity.
> Algorithm 1 is designed as a model-based primal–dual scheme that treats the AMDP solver as a black-box planning oracle. This is intentional: our main contributions are statistical—characterizing the minimax sample complexity in the generative model setting—rather than algorithmic improvements in average-reward planning. Any standard average-reward planner that returns an $O(\varepsilon)$-optimal policy for a given unconstrained AMDP (e.g., LP-based methods, policy iteration, or relative value iteration) can serve as this oracle. If the planner itself is approximate, its error simply appears as an additive planning term in our value gap and can be absorbed into the overall $\varepsilon$ accuracy requirement without changing the leading-order sample-complexity bounds. A detailed comparison of concrete solvers and their computational complexity is therefore orthogonal to the main focus of the paper, but we agree it is an interesting direction for follow-up work and will clarify this perspective in the revised version.
>
> 2.
> We agree that our presentation of Lemma 17 was written too tersely and could easily create confusion. In our general (multichain) setting, the average reward under a fixed policy does depend on the initial state in general. We do not assume a single state-independent gain for all states. Lemma 17 was only meant to capture a high-level intuition: Estimating the span of the bias function $H$ within a constant factor (via an empirical MDP) is a much easier task than learning an $\varepsilon$-optimal policy for the full CAMDP, and therefore does not require the same sample complexity as our main upper bound. This lemma is not used in any essential way in the proofs of the main theorems.
>
> In the revised manuscript we will rewrite Lemma 17, making clear that it is an auxiliary concentration statement for the bias span expressed in terms of our complexity parameters and that no assumption is made that the average reward is constant across states. We will also state that Lemma 17 is not needed for the validity of Theorems 2 and 3, but is included only to convey intuition about why estimating $H$ up to a constant factor is statistically easier than solving the full CAMDP.
>
> We hope these clarifications address the reviewer’s main concerns and we thank you again for the constructive feedback.

---

### Official Review · Reviewer_dT8C · 2025-10-31

**Soundness:** 3
**Presentation:** 3
**Contribution:** 3
**Rating:** 6
**Confidence:** 4

**Summary:**

This paper establishes the first minimax-optimal sample complexity bounds for learning in Constrained Average-Reward MDPs (CAMDPs) under a generative model. The authors propose a model-based primal-dual algorithm that operates under both relaxed feasibility (allowing $\varepsilon$ constraint violation) and strict feasibility (zero violation) settings. The main results show sample complexities of $\tilde{O}(SA(B+H)/\varepsilon^2)$ for relaxed feasibility and $\tilde{O}(SA(B+H)/(\varepsilon^2\zeta^2))$ for strict feasibility, where $\zeta$ is the Slater constant. The paper also provides matching lower bounds, establishing the first minimax-optimal characterization for CAMDPs.

The paper is technically sound with rigorous theoretical analysis:
Strengths:
⦁	The primal-dual algorithm design is well-motivated and properly analyzed
⦁	The proof technique cleverly reduces AMDP concentration to DMDP concentration via Lemma 11
⦁	Both upper and lower bounds are carefully constructed with matching dependencies on key parameters
⦁	The treatment of relaxed vs. strict feasibility is thorough and reveals fundamental differences ($1/\zeta^2$ factor)
Minor concerns:
⦁	The assumption that reward functions $r$ and $c$ are known (lines 184-186) simplifies the problem, though the authors correctly note this doesn't affect leading-order complexity
⦁	Some proof steps rely heavily on prior work (e.g., Vaswani et al. 2022 for discounted concentration), which is acceptable but limits novelty in techniques

Finally, I am familiar with the relevant literature on constrained MDPs and average-reward RL. I carefully checked the main proof techniques and they appear sound, though I did not verify every detail in the appendix. There is a possibility I may have missed some subtle technical issues in the lengthy appendix proofs.

**Strengths:**

Presentation: The paper is generally well-written and organized:
Strengths:
⦁	Clear problem formulation with helpful notation
⦁	Good use of proof sketches in main text with details deferred to appendix
⦁	Figures 1-3 effectively illustrate the hard instance constructions
⦁	The progression from methodology to upper bounds to lower bounds is logical
Areas for improvement:
⦁	The paper is dense with heavy notation that could benefit from a notation table
⦁	Algorithm 1 could be more clearly presented with clearer separation of initialization vs. iteration
⦁	The connection between discount factor $\gamma$ and the transient time/bias span could be explained more intuitively before diving into technical details
⦁	Some key quantities (e.g., the perturbation $\omega$) appear in Algorithm 1 before being properly motivated

Contributions: This is a solid theoretical contribution to constrained RL:
Originality:
⦁	First work to establish sample complexity for CAMDPs in the average-reward setting
⦁	Novel lower bound constructions specific to the constrained setting
⦁	The separation between relaxed and strict feasibility via the $\zeta^2$ factor is new and insightful
Significance:
⦁	Closes an important theoretical gap between unconstrained AMDPs and constrained settings
⦁	The Slater constant $\zeta$ appears naturally and its role is well-characterized
⦁	Results unify understanding across different MDP formulations (finite-horizon, discounted, average-reward)
Limitations:
⦁	Primarily theoretical with no empirical validation
⦁	The gap between relaxed and strict feasibility (factor of $\zeta^2$) is proven tight, but practical implications are unclear
⦁	Limited discussion of when the bounded transient time assumption holds in practice

Strength:
1.	Theoretical completeness: First work to provide matching upper and lower bounds for CAMDPs, establishing minimax optimality with respect to all key parameters ($S$, $A$, $B$, $H$, $\varepsilon$, $\zeta$)
2.	Technical rigor: The proofs are careful and detailed, with novel techniques for handling constraints in the average-reward setting
3.	**Clear problem formulation**: The paper precisely defines two distinct feasibility settings and shows fundamental computational differences between them
4.	Comprehensive treatment: Both weakly communicating and general (multichain) MDPs are addressed with appropriate lower bounds for each class

**Weaknesses:**

1.	No empirical validation: The paper is purely theoretical with no experiments demonstrating the algorithm's practical performance or validating the constants hidden in $\tilde{O}$ notation
2.	Known rewards assumption: While not affecting asymptotic rates, assuming known $r$ and $c$ is somewhat restrictive and simplifies the learning problem
3.	Limited practical guidance:
⦁	No discussion of how to estimate unknown parameters ($B$, $H$, $\zeta$) in practice
⦁	No computational complexity analysis of the algorithm
⦁	Unclear how tight the constants are in practice
4.	Presentation density: The paper packs substantial technical content that may be challenging for readers not deeply familiar with average-reward MDPs and concentration inequalities
5.	Scope limitations:
⦁	Only single constraint case studied (extension to multiple constraints not discussed)
⦁	No discussion of function approximation or large state spaces
⦁	The gap between the generative model setting and online/model-free learning remains unaddressed. It would be valuable to discuss how these theoretical insights might extend to practical applications involving resource-constrained online decision-making, such as LLM inference scheduling under memory constraints [1] or dynamic pricing with capacity limitations [2]
6.	Minor technical gaps:
⦁	The assumption $\varepsilon \in (0, 1/(1-\gamma)]$ appears in theorems without clear justification of why this range is natural
⦁	The relationship between the empirical bias span $\hat{H}$ and true $H$ could be made more explicit.

References
[1] Ao, R., Luo, G., Simchi-Levi, D., & Wang, X. Optimizing LLM Inference: Fluid-Guided Online Scheduling with Memory Constraints. arXiv preprint arXiv:2504.11320.
[2] Ao, R., Jiang, J., & Simchi-Levi, D. Learning to Price with Resource Constraints: From Full Information to Machine-Learned Prices. arXiv preprint arXiv:2501.14155.

**Questions:**

See above.

---

> ### Author Response · Authors · 2025-11-20
>
> We are grateful for the reviewer’s very positive assessment of our work. We are especially encouraged that you regard the paper as a solid and technically rigorous contribution that provides the first matching upper and lower bounds for CAMDPs and clarifies the role of the parameters $(S,A,B,H,\varepsilon,\zeta)$ in constrained average-reward RL. We now address your concerns point by point.
>
>
> 1. No empirical validation.
> Our objective in this paper is purely theoretical: we aim to close a specific information-theoretic gap for constrained average-reward MDPs under a generative model by providing matching upper and lower bounds that are minimax-optimal in all key parameters $(S,A,B,H,\varepsilon,\zeta)$. Achieving such sharp guarantees already requires substantial technical development, and the main contribution of the paper is to fully characterize the fundamental sample complexity of CAMDPs at this level of generality. Empirical evaluation is therefore complementary rather than central to our goals in this work. That said, we agree that it would be interesting to implement the proposed primal-dual scheme and compare different solvers in realistic domains, and we plan to investigate these questions in follow-up work.
>
> 2. Known rewards assumption.
> We agree that assuming $r$ and $c$ are known makes the presentation appear stronger than necessary. Our intention was to follow the standard convention in the generative-model literature: in this setting, learning the transition kernel is statistically harder than learning rewards, and incorporating rewards into the oracle only affects lower-order terms.[1][2] Our upper and lower bounds remain valid (up to constants and logarithmic factors) when $r$ and $c$ are also unknown and sampled together with next states. In the camera-ready version we will add a short remark after the problem statement to clarify this point and explicitly cite these works.
>
> 3. Limited practical guidance and computational complexity.
>
> a. Interpretation and estimation of $B$, $H$, and $\zeta$.
> These quantities are structural properties of the underlying controlled Markov chain. In many common models they admit simple coarse bounds: (i) $H$ is the span of the bias function and can be bounded by the product of the reward range and an effective horizon parameter such as the diameter; (ii) $B$ is a hitting-time bound on reaching recurrent states and is often available from standard Markov-chain arguments or queueing bounds; (iii) $\zeta$ is a feasibility margin determined by how conservatively the constraint threshold $b$ is chosen, and in engineering applications is frequently specified by design.
>
> b. Computational complexity of Algorithm 1.
> Algorithm 1 is a model-based primal--dual scheme: each iteration calls a black-box average-reward MDP solver on a modified reward $r+\lambda_t c$ and then performs inexpensive primal--dual updates. The overall running time is therefore the number of iterations times the cost of solving average-reward planning problems. Any standard solver can be used. These methods have well-known polynomial-time guarantees in the tabular setting. A detailed comparison of concrete solvers and implementations is somewhat orthogonal to the main information-theoretic focus of this paper, so we intentionally keep the algorithmic discussion at this high level. We agree, however, that a more systematic computational study of different planning oracles and implementations of Algorithm 1 would be very interesting, and we see this as a natural direction for follow-up work.
>
> 4. Presentation density.
> We appreciate the comment that the paper is technically dense. Our focus in this work is on developing fairly refined information-theoretic guarantees, which inevitably requires a certain amount of notation and technical machinery. That said, we agree that additional exposition would help readers who are less familiar with average-reward and bias-span techniques. In the revision we will strengthen the high-level introductions around the main results and proofs, provide more intuitive explanations of the key quantities and ideas, and streamline some of the notation, with the aim of making the overall presentation more accessible while keeping the technical content intact.
>
> References:[1] Azar, M. G., Munos, R., and Kappen, H. J. (2013).
> “Minimax PAC Bounds on the Sample Complexity of Reinforcement Learning with a Generative Model.”
> In Proceedings of the 30th International Conference on Machine Learning.
> [2] Sidford, A., Wang, M., Wu, X., Yang, L. F., and Ye, Y. (2018).
> “Near-Optimal Time and Sample Complexities for Solving Markov Decision Processes with a Generative Model.”
> In Advances in Neural Information Processing Systems.

---

> > ### Author Response · Authors · 2025-11-20
> >
> > 5. Scope limitations.
> > We agree that our focus is limited to a single constraint and to the generative-model setting, and that extending the theory to multiple constraints, online interaction, or function approximation would be very valuable. Our goal in this work was to first resolve the cleanest version of the statistical question—minimax-optimal sample complexity for CAMDPs with a generative model. Handling multiple constraints would require a more elaborate vector-valued dual variable and raises new technical challenges for both the upper and lower bounds. Extending the analysis to online, model-free algorithms would additionally require dealing with exploration and non-stationary data. We view these directions as highly promising and, as the reviewer suggests, plan to build on recent works on constrained online decision making and LLM scheduling under resource constraints (e.g., the papers cited by the reviewer) in future work.
> >
> > 6. Minor technical gaps.
> > The restriction $\varepsilon \in (0, 1/(1-\gamma)]$ appears when we convert concentration bounds for discounted MDPs (with effective horizon $1/(1-\gamma)$) back to average-reward guarantees. For larger $\varepsilon$, the problem becomes trivial in the sense that the desired accuracy is worse than the effective horizon.We agree that our discussion of $\hat H$ was too brief. In the revised version we will clarify more explicitly how $\hat H$ relates to the true span bound $H$ and adjust the notation so that this relationship is transparent to the reader.
> >
> > We hope these clarifications address the reviewer’s concerns, and we thank you again for the thoughtful and constructive feedback.

---

> > > ### Comment · Reviewer_dT8C · 2025-11-27
> > >
> > > Thank you for the additional information. I will maintain my rating.

---

### Official Review · Reviewer_c8D3 · 2025-11-04

**Soundness:** 4
**Presentation:** 2
**Contribution:** 3
**Rating:** 6
**Confidence:** 4

**Summary:**

This paper studies the sample complexity of learning in constrained average-reward Markov decision processes (CAMDPs) under the generative model setting. The goal is to determine how many samples are required to compute an \varepsilon-optimal policy that satisfies a long-run average constraint.

Formally, each policy \pi has a steady-state average reward \rho^\pi_r(s) and a constraint value \rho^\pi_c(s), and the learner aims to solve
\max_{\pi}\;\rho^\pi_r(s)\quad \text{s.t.}\quad \rho^\pi_c(s)\ge b.
The analysis distinguishes two regimes: relaxed feasibility (where the constraint can be violated by at most \varepsilon) and strict feasibility (where it must be satisfied exactly).

The authors propose a model-based primal–dual algorithm that iteratively solves empirical unconstrained MDPs of the form r + \lambda c while updating the dual variable \lambda using projected stochastic gradient descent. This construction allows them to derive both upper and lower sample-complexity bounds.

The main results can be summarized as follows:
	•	Relaxed feasibility:
N = \tilde{O}\!\left(\frac{SA(B+H)}{\varepsilon^2}\right)
samples suffice to compute an \varepsilon-optimal policy.
	•	Strict feasibility:
N = \tilde{O}\!\left(\frac{SA(B+H)}{\varepsilon^2\zeta^2}\right),
where \zeta is the Slater constant, measuring the size of the feasible region.
	•	Matching lower bound:
The authors prove \tilde{\Omega}(SA(B+H)/(\varepsilon^2\zeta^2)) for strict feasibility, giving the first minimax-optimal characterization of sample complexity in CAMDPs.

Here S and A denote the number of states and actions, H the span of the bias function, and B a bound on transient time. The results extend prior analyses for (i) unconstrained average-reward MDPs (Zurek & Chen, '24) and (ii) discounted constrained MDPs (Vaswani et al., '22).

The proof builds on Lagrangian duality, strong duality for average-reward MDPs, and confidence bounds for the bias function estimated from the generative model. A Fano-style construction is used to obtain the lower bound. No experiments are provided; this is a purely theoretical paper.

**Strengths:**

1.	Technically solid.
The analysis is mathematically careful and connects several strands of recent work — unconstrained average-reward sample complexity, constrained discounted MDPs, and primal–dual analysis — into one coherent framework.
	2.	Closes a theoretical gap.
The paper provides, for the first time, tight upper and lower bounds for CAMDPs. This completes the theoretical landscape for sample complexity under the generative model.
	3.	Clear separation between regimes.
The dependence on the Slater constant \zeta elegantly captures how strict feasibility increases sample complexity, mirroring known results in convex optimization and constrained RL.
	4.	Methodologically sound.
The primal–dual approach is principled and aligns well with the structure of constrained reinforcement learning.

**Weaknesses:**

1.	Difficult presentation.
The exposition is unnecessarily dense. Key symbols are introduced before being defined (\tilde{c}, M’, \hat{M}), and the algorithmic steps (e.g., the reason for reward perturbations or the \epsilon-net projection for \lambda) are not well motivated. The reader has to reconstruct much of the reasoning from context.
	2.	Incremental novelty.
The core ideas extend known results from discounted CMDPs and unconstrained AMDPs rather than introducing new analytical techniques. The adaptation to the average-reward case is nontrivial but conceptually straightforward.
	3.	Limited intuition.
The paper would benefit from more discussion of how the constants H, B, and \zeta influence learning difficulty, or how they compare to analogous quantities in the discounted setting. At present, the results are formal but not deeply interpretable.
	4.	Limited to no empirical or illustrative validation.
Even a small numerical experiment illustrating the scaling with \zeta or comparing relaxed and strict feasibility would have improved readability and intuition.
	5.	Accessibility.
The technical writing assumes significant familiarity with the literature on bias-span bounds and average-reward MDPs. Without this background, it’s very hard to follow.

**Questions:**

1.	Novelty relative to Vaswani et al. (2022) and Zurek & Chen (2024).
	•	What is genuinely new in the analysis beyond extending those frameworks to the average-reward case?
	•	Are there any technical hurdles specific to the average-reward setting (e.g., lack of contraction) that required new ideas?
	2.	Understanding the constants.
	•	Can you provide intuition for how the bias-span H, transient time B, and Slater constant \zeta control the complexity?
	•	In practice, how might one estimate or bound these quantities?
	3.	Algorithm design.
	•	Why is reward perturbation needed in the primal–dual algorithm?
	•	How sensitive is the performance or convergence to the step size and to the discretization of the dual variable?
	4.	Broader applicability.
	•	Can your results be extended to settings without a generative model — e.g., online learning or policy-gradient approaches?
	•	Would similar rates hold if only sample trajectories were available?
	5.	Clarity and structure.
	•	Consider reorganizing the exposition so that the notation is introduced before use, and include a brief intuitive overview of the proof strategy. This would substantially improve readability.

---

> ### Author Response · Authors · 2025-11-20
>
> We thank the reviewer for the careful reading and the constructive comments.
> We are glad that you find the work technically solid, that it closes a theoretical gap for constrained average-reward MDPs, and that the dependence on the Slater constant $\zeta$ and the use of a primal--dual approach are methodologically sound.
> Below we address your concerns point by point.
>
>
> 1. Novelty (Question 1)
>
> a. Zurek & Chen (2024) obtain near-minimax sample-complexity bounds for unconstrained average-reward MDPs under a generative model, characterized by $H$ and $B$.
> Vaswani et al. (2022) analyze discounted constrained MDPs using a primal--dual scheme.
> Neither work treats constrained average-reward problems, and no prior result provides tight upper and lower bounds for CAMDPs under a generative model. Our paper precisely closes this gap.
>
> b. Adapting the discounted primal-dual framework to average reward is not a mechanical change of variables. We cannot directly exploit the Bellman contraction.
>   Instead, we leverage the structure of bias functions and transient time bounds, bounding the deviation between discounted values $V_\gamma^\pi$ and average gains $\rho^\pi$ in terms of $H$ and $B$.
>
> c. Our lower-bound construction introduces a new multichain CAMDP architecture with carefully designed constraint rewards that encode the feasibility margin $\zeta$. Designing this architecture so that it simultaneously controls the bias span, transient time, and feasibility gap is technically delicate. By embedding $\zeta$ into the transition-reward structure, we obtain a family of CAMDPs whose optimal policies differ only within a narrow constraint gap, forcing any algorithm to distinguish them with $\Omega(\zeta^{-2})$ effort.
> To the best of our knowledge, this is the first information-theoretic lower bound for constrained average-reward RL that explicitly captures the role of the Slater margin $\zeta$.
>
>
> 2. Understanding the constants (Question 2)
>
> a. Bias-span $H$:
> $H$ is the span of the bias function.
> Intuitively, it measures how ''uneven'' the long-run relative values of states are:
> if some states are much better than others, then the algorithm must accurately estimate
> small differences in long-term averages, which is statistically harder.
> Such bounds are standard in average-reward theory and can often be obtained from domain knowledge.
>
> b. Transient time $B$:
> $B$ upper-bounds the expected time needed to reach the recurrent class under any stationary policy.
> Large $B$ means the system can spend a long time in transient regions whose contribution to the average reward and the constraint is difficult to estimate from finite samples.
> From a complexity point of view, this behaves like an additional horizon: before the chain enters its steady-state regime, the learner must track rewards and constraints over a transient window of length up to $B$.
>
> c. Slater constant $\zeta$:
> $\zeta $ is the slack that the best feasible policy has relative to the constraint.
> When $\zeta$ is small, there exist policies whose long-run constraint values are extremely close to the threshold $b$.
> Any algorithm that must output a strictly feasible policy then has to distinguish between policies with constraint values that differ by at most $O(\zeta)$.
>
> d. How to bound them in practice.
> In many applications we can use coarse structural bounds:
>    $H$ can be bounded by the product of the reward range and a mixing-time or diameter parameter of the MDP;
>   $B$ can be bounded by a hitting-time bound on reaching recurrent states;
>  $\zeta$ is often known or controllable, since it is determined by how conservatively the constraint $b$ is chosen.
>
>
> 3. Algorithm design: reward perturbation and dual discretization (Question 3)
>
> a. Why reward perturbation? Our use of reward perturbation is mainly to align our setting with the non-asymptotic analysis of Vaswani et al. (2022), whose results we invoke in the final step of the proof. Their key structural theorems are stated under a small random perturbation that guarantees uniqueness and stability of optimal policies. Adopting the same perturbation in our AMDP–DMDP reduction lets us use their bounds as a black box instead of re-developing the stability and concentration machinery for constrained average-reward MDPs. We refer interested readers to Vaswani et al. and would be happy to elaborate during the discussion phase.
>
> b. Step size and discretization. Our theoretical choices $\eta=\sqrt{U/T}$ and grid resolution $\varepsilon_1$ follow the mirror-descent regret and covering-number arguments used by Vaswani et al. and lead to Theorem 1. In practice, moderate changes in $\eta$ only affect convergence speed, not the final performance, and one can use a continuous projection for $\lambda$ instead of an explicit $\varepsilon$-net; the discretization is mainly a proof device to obtain uniform regret bounds over $\lambda\in[0,U]$.

---

> > ### Author Response · Authors · 2025-11-20
> >
> > 4. Broader applicability beyond the generative model (Question 4)
> >
> > Our results are stated in the generative-model setting because:
> >  It is the standard oracle model for sample-complexity questions in modern RL theory (including Zurek \& Chen), and it cleanly separates statistical from exploration issues.
> >  Our lower bounds are information-theoretic for this strong oracle, so they automatically apply to weaker interaction models (online learning from a single trajectory, policy-gradient methods, etc.).
> >
> > Extending the upper bounds to online interaction (where the agent must both explore and control constraints) is an interesting direction that we see as complementary to this work.
> > The structural insights we obtain---especially the explicit dependence on $H$, $B$, and $\zeta$, and the primal-dual reduction to discounted unconstrained MDPs---suggest that analogous guarantees should be attainable in optimistic model-based or policy-gradient schemes, but a full treatment would require additional algorithmic machinery and is beyond the current scope.
> >
> > 5. Clarity and structure. (Question 5)
> >
> > a. We acknowledge that the current exposition is dense and that some symbols (e.g., $M'$, $\hat M$) appear before being fully motivated.
> > This is something we can and will improve.
> > In the revision we will move the definitions of $M'$ and $\hat M$ to Section 3 before they are first used, and will explicitly summarize the three levels of MDPs we use:
> > the true CAMDP $M$;
> > the empirical CMDP $\hat M$ induced by the generative model;
> > and the reference CAMDP $M'$ used in the primal-dual analysis.
> >
> >
> > b. An overview of proof strategy.
> >
> > (i) Primal--dual guarantee on a reference CAMDP.
> > We first analyze Algorithm 1 purely at the optimization level.
> > Running the primal--dual updates on the empirical CMDP $\hat M$ induces a sequence of unconstrained average-reward MDPs (AMDPs) with modified rewards $r_p + \lambda_t c$.
> > Using a standard mirror-descent regret argument, we show that the averaged policy produced by the algorithm is nearly optimal for a carefully defined reference CAMDP $M'$ built on the empirical model.
> > This step yields a value gap between our policy and the optimal policy of $M'$.
> >
> > (ii) Relating $M$, $M'$, and $\hat M$ via AMDP concentration. Second, Lemma 9 decomposes the suboptimality of our policy on the true CAMDP $M$ into three pieces: the optimization error on $M'$ plus two statistical errors coming from the discrepancy between $M$ and $\hat M$. These statistical terms can be written as a collection of concentration bounds for a family of unconstrained AMDPs corresponding to different dual parameters $\lambda$. In other words, the only remaining task is to control the deviation between empirical and true average rewards in these AMDPs under the generative model.
> >
> > (iii) From AMDP to discounted MDP and application of Vaswani et al. (2022)
> > Finally, we invoke Lemma 11 that converts each AMDP concentration term into an analogous concentration bound for a suitably chosen discounted MDP (DMDP) with discount factor $\gamma$.
> > This allows us to apply the analysis of Vaswani et al. (2022) for perturbed discounted CMDPs.
> > Combining their discounted bounds with our AMDP-DMDP reduction yields high-probability control of all the statistical error terms, and thus the sample-complexity guarantees stated in Theorems 2 and 3.
> >
> > We hope that these clarifications and planned revisions address the reviewer’s concerns.
> > Thank you again for the thoughtful feedback.

---

### Meta-Review · Area_Chair_gxJK · 2026-01-06

**Summary:**

This paper studies RL in the constrained average-reward setting assuming a generative model of the environment is available. The learner's performance is the sample complexity of learning a near-optimal policy. A key contribution here is a model-based primal-dual algorithm that admits sample complexity bounds under two relevant regimes (relaxed and strict feasibility). The bound for the strict feasibility is minimax-optimal, as confirmed the lower bound derived in the paper.

The reviewers unanimously agree that the paper addressed an interesting and important problem in RL, and makes solid and technically rigorous contributions to the field by closing a gap in a technically challenging and practically relevant RL problem. The rebuttal adequately addressed the questions -- technical and presentational -- raised by the reviewers. Therefore, I recommend acceptance.

**Reviewer Concerns:**

__Concerns related to technical aspects.__ The reviewers raised questions and concerns related to technical aspects that include: (1) limitations of generative model and extension to more complex RL settings; (2) extensions beyond single constraints; (3) extension beyond tabular setting, limitations of the model-based approach to incorporate function approximation; (4) the impact of approximation error due an inexact oracle; .

I do believe that the rebuttal adequately and precisely addressed all these points, many of which are to addressed in future work --and I fully agree with the authors.

__No empirical evaluation.__ The paper lacks an empirical evaluation, as mentioned by the reviewers. Even simple numerical experiments could be a nice addition to show case practicality of the proposed methods. Nevertheless, considering that it's a purely theoretical paper, this issue is fine.

__Limited novelty.__ There were some questions regarding the novelty of the presented algorithm and analyses considering methods for similar settings (e.g., discounted setting or unconstrained average-reward), although the reviewers who raised these were readily positive. As the rebuttal clarified --and I agree--, deriving tight sample complexity bounds renders challenging and non-trivial. Further, the sample complexity lower bound, which is key contribution and a nice addition to the literature, justifies well the level of technical novelty in the paper.

__On the intuition of sample complexity bounds.__ There were questions regarding interpretability of parameters appearing in the sample complexity bounds. These were fully addressed and clarified in the rebuttal. The inclusion of discussion in the paper will further improve the presentation.

**Reviewer Scores:**

- Reviewer c8D3: Their raised questions were addressed well in the rebuttal. So I think the reviewer would maintain the score.
- Reviewer dT8C: They stated that they maintain the score.
- Reviewer 4Q3K: Their raised concerns were sufficiently and adequately addressed in the rebuttal. So, I see it likely that the reviewer would increase.
- Reviewer 6HuV: They stated that they maintain the score.

---

### Decision · Program_Chairs · 2026-01-26

Accept (Poster)